# Single strain control of microbial consortia

Alex J. H. Fedorec [1✉], Behzad D. Karkaria [1], Michael Sulu [2] & Chris P. Barnes [1,3✉]

The scope of bioengineering is expanding from the creation of single strains to the design of microbial communities, allowing for division-of-labour, specialised sub-populations and interaction with "wild" microbiomes. However, in the absence of stabilising interactions, competition between microbes inevitably leads to the removal of less fit community members over time. Here, we leverage amensalism and competitive exclusion to stabilise a two-strain community by engineering a strain of *Escherichia coli* which secretes a toxin in response to competition. We show experimentally and mathematically that such a system can produce stable populations with a composition that is tunable by easily controllable parameters. This system creates a tunable, stable two-strain consortia while only requiring the engineering of a single strain.

[1] Department of Cell and Developmental Biology, University College London, London, UK. [2] Department of Biochemical Engineering, University College London, London, UK. [3] UCL Genetics Institute, University College London, London, UK. ✉email: alexander.fedorec.13@ucl.ac.uk; christopher.barnes@ucl.ac.uk

Techniques for the assembly[1] and synthesis[2] of DNA sequences have enabled the construction of complex, very large, even chromosome scale[3], synthetic biological systems. Recent efforts to systematically characterise genetic "parts"[4,5] and develop software tools to produce DNA sequences from function specification[6], are enabling synthetic biological systems in which large numbers of regulatory proteins and promoters are involved. The production of each of these proteins sequesters resources that would otherwise be used by the host organism for growth and therefore leads to a reduction in growth rate[7,8]. This provides a selective advantage to loss-of-function mutants that arise, suggesting that the functional period for the increasingly complex synthetic biological systems of the future will become ever more ephemeral.

Attempts have been made to minimise burden[8,9], make mutation deadly[10], or periodically displace mutating populations with a functioning population[11]. Over the past decade, there have been attempts at division-of-labour, in which a system is split into subcomponents and distributed into specialised sub-populations of cells[12,13]. This minimises the burden that is placed on individual cells which reduces, but does not remove, the selective advantage of loss-of-function mutations. In addition, the creation of synthetic communities allows the diversification and compartmentalisation of functions, modularisation, spatio-temporal control and mechanisms for biosafety[14].

Beyond the benefits for biotechnology applications of division-of-labour, synthetic biological systems will be increasingly applied in environments in which microbiomes are already present, such as for bio-therapeutics[15] or bio-remediation[16]. In these settings, it is often undesirable to exterminate the resident species other than to remove pathogens. However, non-native species can find it difficult to persist long enough to be useful[17]. The safety, efficacy, and societal acceptance of engineered bacteria in these environments rely on their predictable function[18], requiring predictable persistence and population dynamics. As an example, butyrate production by probiotic bacteria has been demonstrated to be beneficial for gut health[19], but over production of butyrate may be harmful[20]. To sustain butyrate production within a healthy window a butyrate-producing bacterial strain would need to be maintained at a consistent level. Another example is in the use of biosensing strains to detect and report on disease states in vivo[21]. Without knowledge of the size of the reporting population, it is impossible to distinguish between a weak signal from a large population and a strong signal from a small population.

The fundamental challenge with constructing, or integrating into, such heterogeneous communities is the principle of competitive exclusion, which states that two species competing within the same niche cannot coexist[22]. The principle should, perhaps, include the caveat: "in stable environments and in the absence of other interactions", which may help to explain supposed deviations such as the "paradox of the plankton"[23]. Wild bacteria live in complex communities[24] with mutualistic and competitive interactions producing complex dynamics[25]. Previous attempts to design synthetic microbial communities have relied on spatial segregation[26] or mutualism[27] to maintain multiple sub-populations. The control of the density of monocultures has been achieved through the use of quorum sensing to control self-killing[28] and recently this has been extended to a two-species system[29]. More complex predator-prey systems have also been developed that produce oscillatory populations of two strains[30]. These systems involve the engineering of all strains within the community. However, this requirement may not be desirable in industrial settings, in which strains may have already been optimised for a particular function, and is clearly not possible when working in natural environments such as the human gastrointestinal tract.

In order to control a community through a single constituent, we require a mechanism that allows control of the growth rate of one or more competitors at a distance. Here we suggest bacteriocins, secreted antimicrobial peptides, as such a mechanism and detail the construction and characterisation of a control system that uses them. A wide range of bacteriocins are produced in natural microbial communities such as the human intestinal microbiota[31] where they have an important role in niche competition[32]. The majority of bacteriocins target species that are closely related to the producing species, though some broad-spectrum bacteriocins exist[33]. The modular nature of bacteriocins has been exploited to alter[34] and expand[35] their spectrum of activity. Many bacteriocin expressing species encode several different bacteriocins in their genome[36]. We have previously demonstrated the ability of the bacteriocin microcin-V to improve plasmid maintenance[37]; a challenge that includes preventing competitive exclusion. Further, gram-positive bacteriocins have been used to produce commensal and amensal interactions along with all pairwise combinations of the two[38]. By limiting ourselves to the engineering of a single strain, this system could be repurposed for applications in different settings by selecting the bacteriocins appropriate for the target niche.

In this work, we present the development of a strain of bacteria that is able to control a co-culture by the expression and secretion of a bacteriocin that targets a competitor. We engineer the strain to repress bacteriocin expression in response to a quorum molecule and ascertain environmental conditions that affect co-culture dynamics. The system is made self-regulating by enabling the engineered strain to produce the quorum molecule, and we show mathematically and experimentally the stabilisation of a co-culture. Finally, using a computational approach, we suggest improvements to the engineered system to allow it to function more robustly.

## Results

**Bacteriocins enable population control at a distance**. We start with a consortia consisting of a faster-growing *E. coli* MG1655 competitor strain, and a slower-growing *E. coli* JW3910[39] engineered strain which can overcome competitive exclusion by secreting the bacteriocin (an antimicrobial peptide) microcin-V[37] (Fig. 1a). The engineered strain is further weakened through the constitutive expression of mCherry from one of the plasmids (p63_AF043), which also allows the separation of the engineered and competitor strains in the plate reader[40]. The competitor strain excludes the engineered strain when the bacteriocin is mutationally inactivated[41], but is outcompeted with complete bacteriocin expression (Fig. 1b).

When we look more closely at the population dynamics of a co-culture, we observe an initial phase of competitive exclusion before the killing of the competitor occurs (Fig. 2a, b). The time it takes before the engineered strain outcompetes the competitor (the point at which the proportion of the population constituted by the engineered strain is greater than its initial proportion) is related to the initial population ratio (Fig. 2c). This suggests that a simple control strategy would be to passage more frequently if the engineered strain is dominating or less frequently when the competitor dominates.

**Restricting growth favours bacteriocin producers over faster growers**. Using a simple mathematical model (Supplementary Information), we are able to simulate growth dynamics and bacteriocin production in a chemostat environment. Although our experimental set-up does not afford a true chemostat environment, the computational simulation shows that repeated dilution of batch cultures produces comparable outcomes to a

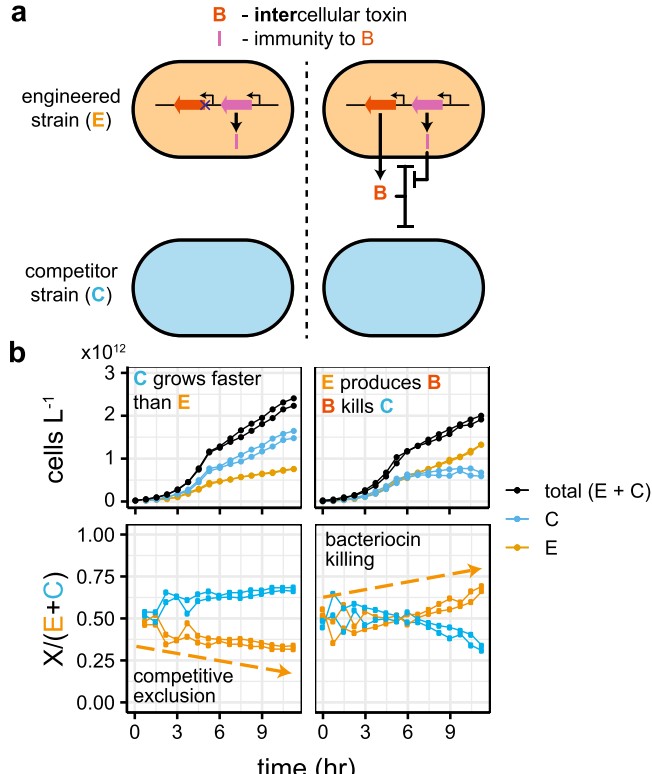

**Fig. 1 Bacteriocins can overcome competitive exclusion. a** The engineered strain with inactivated *cvaC* gene produces no bacteriocin. With an intact gene, the strain secretes a bacteriocin, which kills the competing strain. **b** Co-cultures of the engineered and competitor strain were grown in a microtitre plate. With mutationally inactivated bacteriocin, the faster-growing competitor excludes the engineered strain. Bacteriocin secretion leads to the killing of the competitor and domination by the engineered strain. Each panel includes two replicates, with each point showing the value of individual replicates calculated from plate reader measurements.

chemostat, as long as the period between dilutions is not too long; 2 h or less for our conditions (Supplementary Fig. 6). Analysis of our model shows that, depending on its starting composition and dilution rate, the co-culture will tend towards the exclusion of one or the other strain (Fig. 3a). This reinforces our previous statement that, with control of dilution rate, one could switch between states in which the engineered strain or the competitor strain dominates based on the current community composition. Methods to achieve this have previously been shown using nutrient control rather than dilution rate[42].

Although we do not directly control the dilution rate of our co-cultures, we can emulate the impacts of dilution rate on population density, growth substrate and bacteriocin concentration by serially diluting co-cultures and observing their dynamics at a range of initial densities (Fig. 3b). In the most dilute cultures, the initial concentration of bacteriocin is very low and the growth substrate is plentiful, allowing the competitor strain to dominate the population before the engineered strain is able to get a foothold. As the initial density increases so too do the initial bacteriocin concentration and the number of engineered cells secreting bacteriocin. At the same, the initial substrate concentration is reduced, narrowing the window over which the competitor can outcompete the engineered strain. This combination leads to a shortening of the period of competitive exclusion and an increase in the rate at which the engineered strain takes over (Fig. 3b). The same experiment is performed at a range of initial co-culture ratios (Fig. 3c and Supplementary Fig. 1). We see that, as the initial density increases, the system moves from favouring the competitor to favouring the engineered strain, just as we showed by changing the dilution rate with our mathematical model (Fig. 3a).

The death rate of the competitor strain reduces after the cultures reach a peak density; this is most notable at the highest initial density shown in Fig. 3b. This demonstrates a potential point of deviation between the mathematical model, in which cells are in a chemostat and therefore growing exponentially, and our plate reader experiments in which stationary phase is reached. The 5 h sample time point chosen for Fig. 3c and later

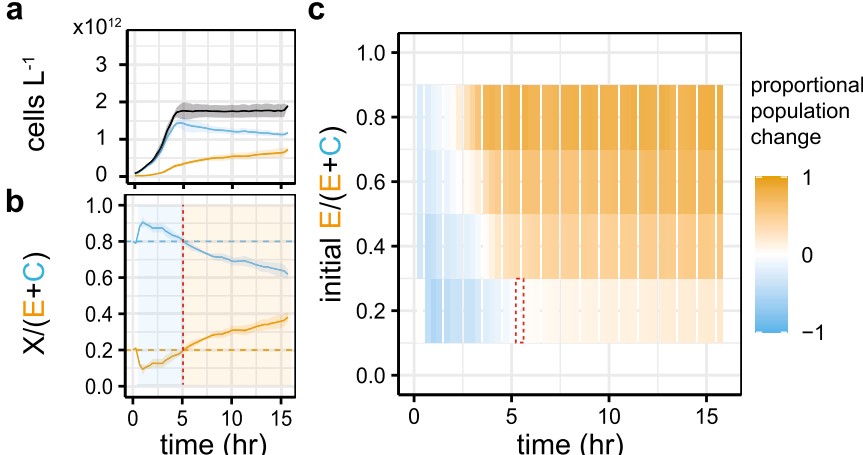

**Fig. 2 Larger engineered populations kill competitors faster. a** The two strains are co-cultured in a microtitre plate and the subpopulation fractions are determined using fluorescent proteins in the engineered strain and the FlopR software[40], allowing us to track the population dynamics over time. **b** The blue shaded area shows the period during which the competitor ratio is above its initial ratio; the orange shaded area shows the same for the engineered strain. The red dashed line indicates the time point at which the change from competitor to engineered strain dominance occurs. **c** Graph showing the "winning" strain at each timepoint of co-cultures with four different initial ratios. The colour and saturation of each block are given by the difference between the initial strain ratio and the ratio at that timepoint. The dashed red box shows the point in **b** at which the engineered strain begins to win. In **a** and **b**, lines show the mean of the rolling mean (centred window size = 3) and the shaded ribbons show the standard deviation of three replicates. In **c**, each horizontal summarises three replicate timecourses; the shaded rectangles show the mean proportional population change at each time point.

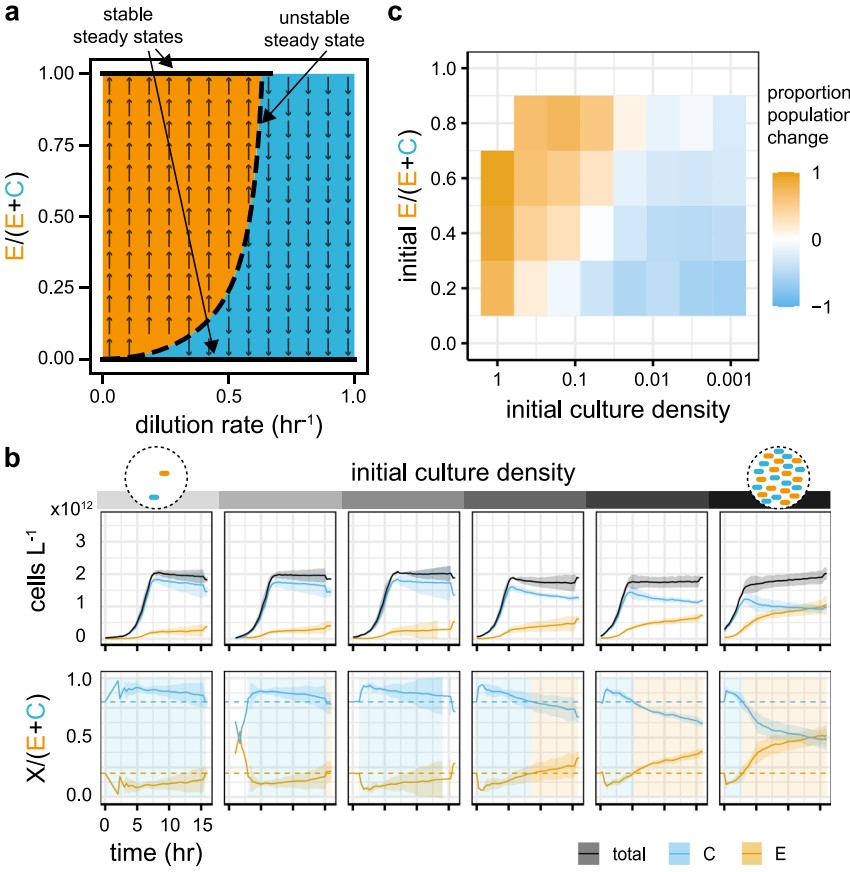

**Fig. 3 Environment dilution can control relative strain fitness. a** Analysis of the mathematical model with varying dilution rate. Solid black lines show the stable steady states (extinction of one or other strain), the dashed black line indicates an unstable steady state. The arrows show the direction of population change on either side of the unstable manifold. **b** Dilution rate is approximated by varying the initial density of the co-cultures in the microtitre plate; lower initial density approximating faster dilution rates. We observe the same switch, from competitive exclusion to bacteriocin killing, predicted by the model. **c** The "winning" strain after 5 h of growth in co-culture, over a range of initial population ratios, mimics the model prediction. In **b** the lines show the mean of the rolling mean (centred window size = 3) and the shaded ribbons show the standard deviation of three replicates. In **c**, each shaded rectangle shows the mean proportional population change of three replicates, calculated at 5 h post-inoculation.

figures is approximately at the end of the exponential growth phase and before the change in killing rate occurs.

**Flipping competitive advantage via an exogenous inducer**. For some applications, total control of the environmental parameters such as dilution rate is unrealistic. As such, we need another mechanism for switching state from one strain to the other dominating, which we achieve through control of the bacteriocin production rate using N-3-oxohexanoyl-homoserine lactone (3OC6-HSL) (Fig. 4a, b). Using an agar spot inhibition assay, we demonstrate exogenous control of killing in a dose-dependent manner (Fig. 4c, d).

Our mathematical model demonstrates that, at lower rates of bacteriocin production, the killing of the competitor is reduced, allowing it to dominate, whereas at higher bacteriocin production rates the engineered strain dominates (Fig. 5a). Co-cultures grown in various concentrations of 3OC6-HSL demonstrate that as the concentration of 3OC6-HSL increases, the killing of the competitor by the engineered strain is relieved and the system moves from engineered strain domination to competitor domination (Fig. 5b and Supplementary Fig. 2). This is reflected in the changes that we see after 5 h of competition (Fig. 5c). However, if the population starts with a low or high proportion of engineered cells, changing 3OC6-HSL concentration is not able to flip which strain has the competitive advantage, limiting the communities that we are able to have an effect on. This is due to

the maximal fold change in the expression level of the bacteriocin being ~10 (Fig. 4d) in the system as implemented. Using the data from Fig. 5c, we can overlay the possible regions for minimal and maximal bacteriocin production rate on Fig. 5a; the extreme ends of the regions is ~10 fold difference in bacteriocin expression. Ultimately, increasing the fold change in bacteriocin expression will increase the range of consortia composition over which our system can operate.

**Autonomous community regulation**. The choice of 3OC6-HSL for control of bacteriocin expression allows us to use population density for control by adding the expression of the 3OC6-HSL synthase gene to the engineered strain (Fig. 6a). This affords the creation of an autonomous system rather than one that needs environmental control[29,43]. This system should then be able to sense and respond to competitive exclusion through changing concentrations of 3OC6-HSL (Fig. 6b). The addition of an arabinose inducible promoter driving expression of the 3OC6-HSL synthase gene gives us the ability to tune the production rate of 3OC6-HSL per cell; higher levels of arabinose induce 3OC6-HSL synthase expression more strongly. By having a higher concentration of 3OC6-HSL per engineered cell, the threshold for turning off the bacteriocin is met at a lower population density.

The system was constructed using a modular approach, with fluorescent proteins cloned downstream of each promoter, which

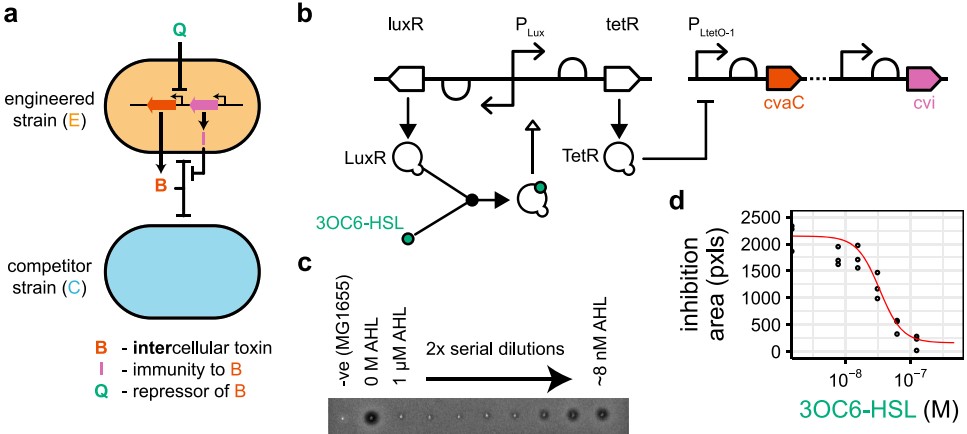

**Fig. 4 Exogenous control of bacteriocin production. a** The expression of bacteriocin can be controlled by the addition of a quorum molecule ($Q$ = 3OC6-HSL) into the environment. **b** The system is spread across two plasmids. 3OC6-HSL binds to LuxR, LuxR-AHL induces the expression of TetR, which represses bacteriocin expression. **c** Conditioned media from the growth of the engineered strain in a range of 3OC6-HSL concentrations was spotted onto a lawn of EcM-Gm-CFP (Supplementary Information) leading to zones of inhibition from bacteriocin killing. **d** The inhibition areas are quantified using an image processing pipeline and used to fit a Hill function for bacteriocin expression. The image in **c** shows a representative slice from a larger plate containing three replicates (Supplementary Fig. 3). In **d**, points show values of three replicates. The red line shows the fitted Hill function.

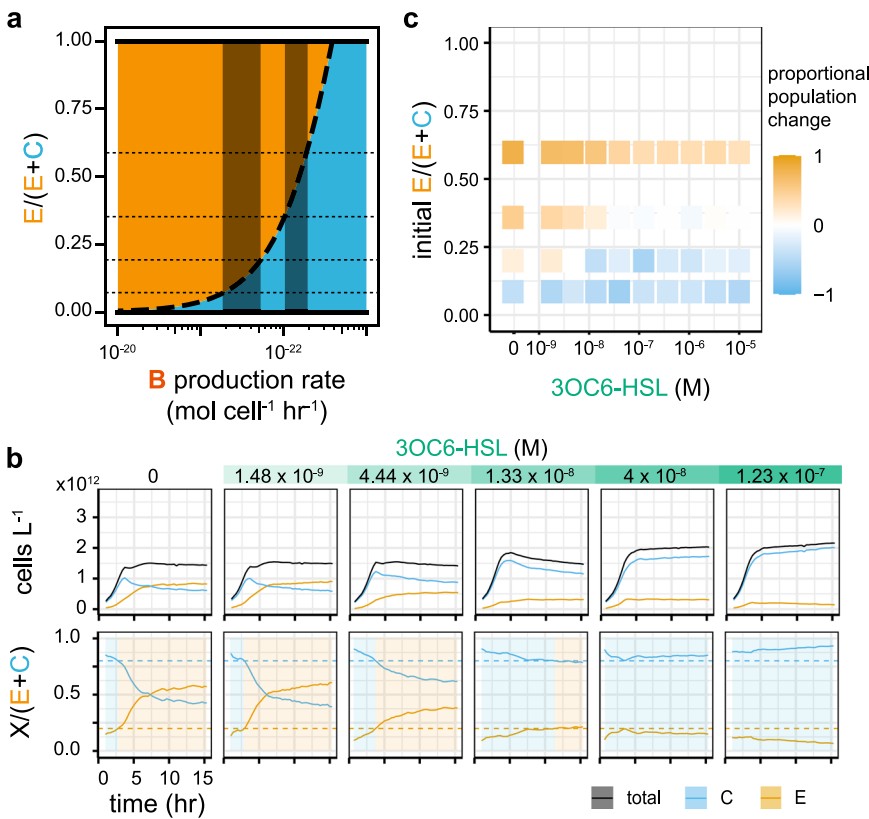

**Fig. 5 Exogenous control of competition. a** Analysis of the mathematical model with varying bacteriocin expression rate. Note that the *x* axis is reversed to mimic the direction of 3OC6-HSL control, i.e., high bacteriocin production rate on the left, low bacteriocin production rate on the right. The dotted horizontal lines mark the initial population ratios from **c**. The shaded regions show the possible regions of maximum and minimum bacteriocin production rate based on **c**. **b** Dynamics of co-cultures grown in different concentrations of 3OC6-HSL. **c** The "winning" strain after 5 h of co-culture, with varying 3OC6-HSL concentration, across several initial population ratios. Mapping these results onto **a** shows the possible regions of minimal and maximal bacteriocin expression rate within which our system is operating. In **b**, the lines show the rolling mean (centred window size = 3) of one replicate. Further replicate shown in Supplementary Fig. 2. In **c**, each shaded rectangle shows the proportional population change of one replicate, calculated at 5 h post-inoculation.

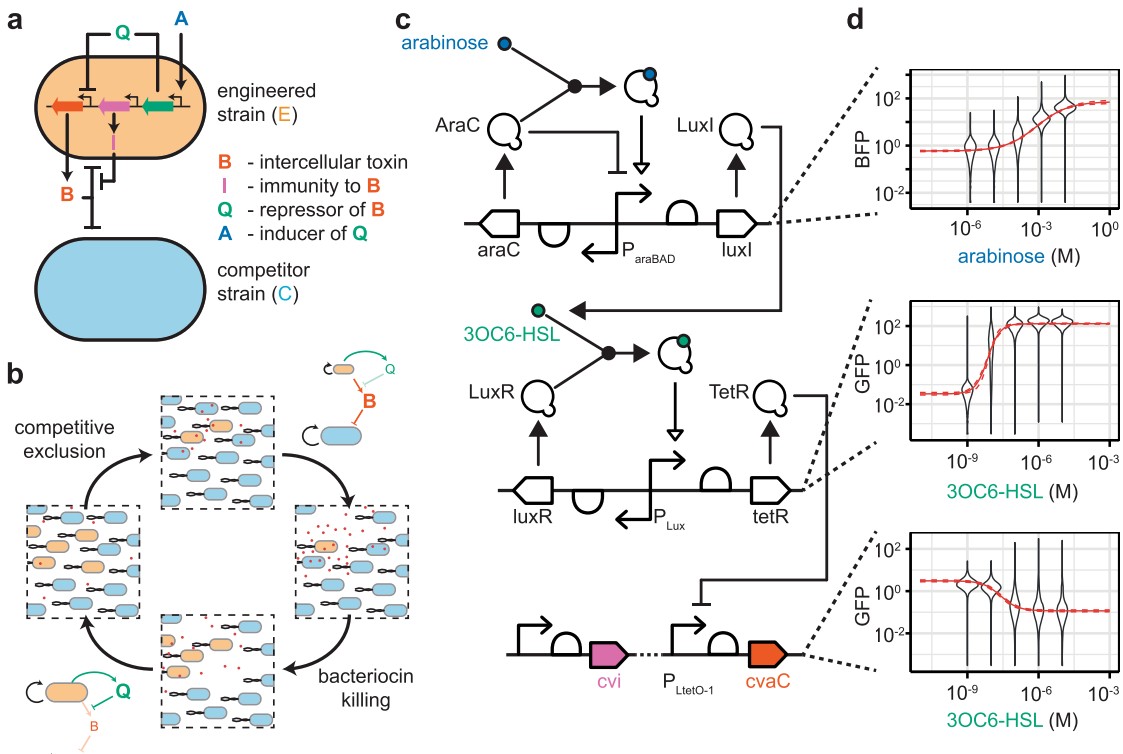

**Fig. 6 Autonomous control of bacteriocin production. a** The engineered strain produces its own quorum molecule at a rate that is tunable by arabinose induction. **b** Bacteriocin production is turned off when the engineered population density is high, allowing the competitor to grow, and competitive exclusion to complete the cycle. **c** The system was constructed in three parts: arabinose inducible expression of LuxI, 3OC6-HSL inducible expression of TetR, and TetR repressible expression of microcin-V. **d** Each part was characterised by cloning a fluorescent reporter protein (GFP or BFP) downstream of the relevant promoter and measuring expression using flow cytometry. Hill functions were fitted to the data using a Bayesian method. Violins show the distribution of all flow cytometry events from three replicates. The solid lines = mean prediction, dashed lines = 95% credible region in the mean.

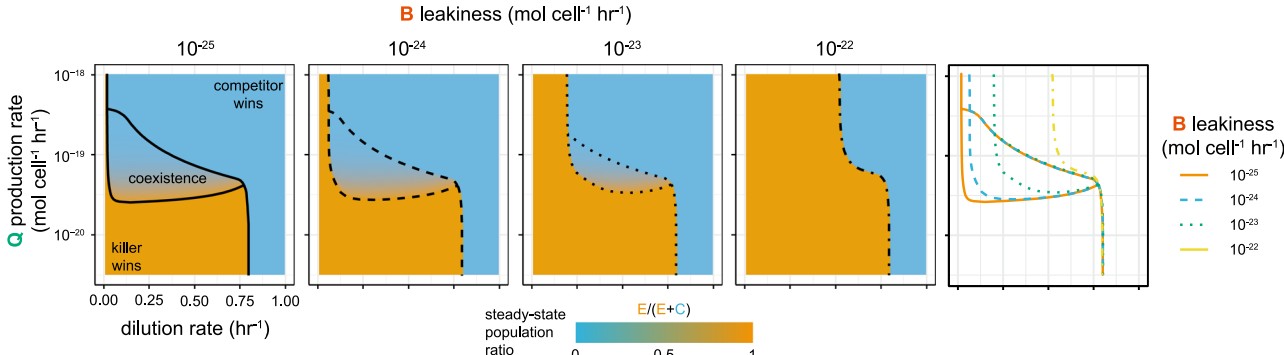

**Fig. 7 Bacteriocin leakiness reduces the region of co-existence.** Simulations of the parametrised mathematical model show regions in which stable co-existence is achieved. As the leakiness of the bacteriocin expression increases, the region shrinks until it eventually disappears. The gradient within the region also shows that the steady-state ratio of engineered strain to competitor strain is dependent on the quorum molecule production rate, which is controllable through arabinose induction.

allowed us to test the function of each component as we progressed (Fig. 6c, d) and enabled us to use modelling during the construction and characterisation to determine whether the system, as constructed, could achieve the stable co-existence that we desired. The copy number of the plasmid bearing the control elements (arabinose inducible LuxI and 3OC6-HSL inducible TetR) was explored (Supplementary Fig. 4) to ensure the maximal expression range of TetR, inferred from GFP expression, under varying arabinose concentrations without overburdening the host cells.

A steady-state analysis of this system shows regions of stable co-existence which vary in size depending on the leakiness of

bacteriocin expression (Fig. 7). Reducing the leakiness enlarges the region of stable co-existence, allowing stability at lower dilution rates and over a greater range of quorum molecule production rates which would improve the robustness of co-existence. Importantly, changing the production rate of 3OC6-HSL, which we can do through arabinose induction of LuxI, allows us to move the system into a region with stable co-existence. In addition, we can tune the population ratio within the stable co-existence region by using the 3OC6-HSL production rate, with a higher rate resulting in a lower fraction of engineered strain in the community. As the leakiness approaches the maximal expression rate (i.e., the bacteriocin cannot be turned

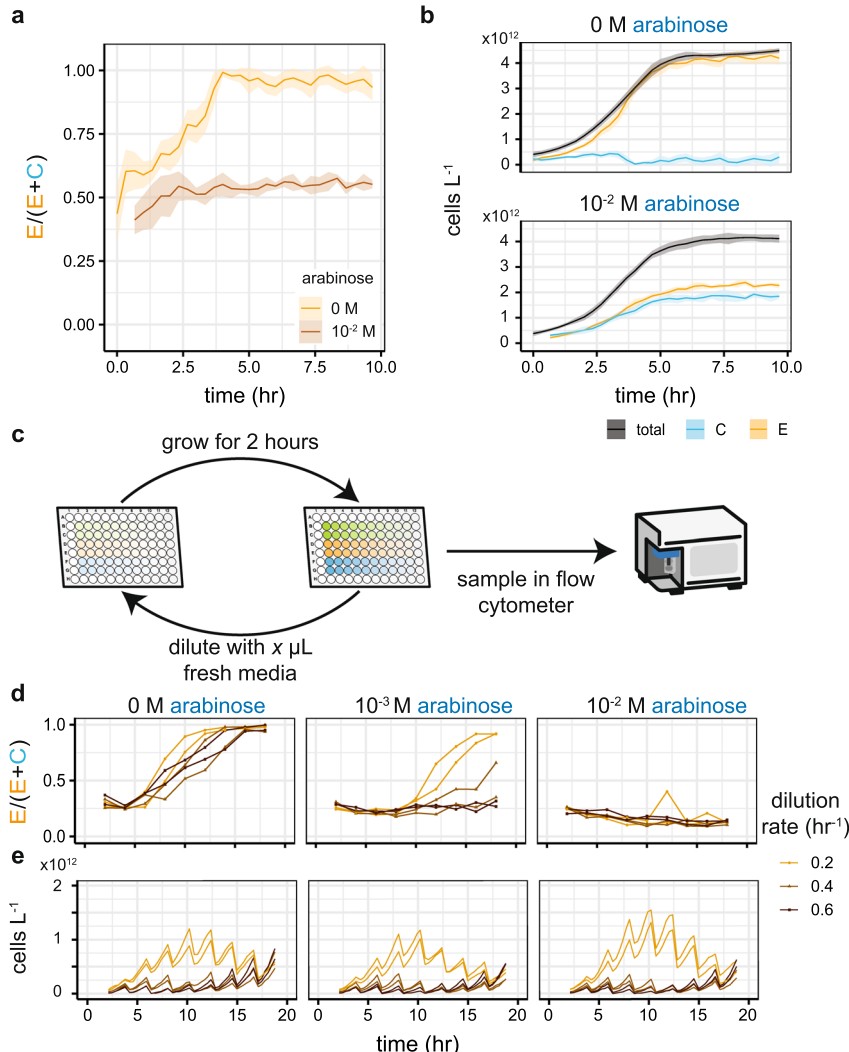

**Fig. 8 Demonstration of community stabilisation. a** Co-cultures of engineered and competitor strains grown in microtitre plates. Without arabinose in the media, no 3OC6-HSL is produced and the bacteriocin is not switched off, resulting in the competitor going extinct. With 10 mM arabinose, we observe stable co-existence. **b** Growth curves and inferred subpopulations from **a**. The cultures reach saturation after ~5 h. **c** Co-cultures in microtitre plates were repeatedly grown and diluted with samples taken for analysis by flow cytometry. A range of dilution volumes was used to approximate chemostat dilution rates. **d** Population ratios were attained by clustering on the flow cytometry data. Without arabinose, the competitor strain goes extinct, though this occurs faster at the lowest dilution rate. At 10 mM arabinose, we observe co-existence at all dilution rates, for the period of the experiment. At 1 mM arabinose, the competitor is close to extinction at the lowest dilution rate and appears to be outcompeted at the intermediate dilution rate. **e** Growth curves demonstrate the dilution dynamics and show that saturation is not reached under any of the conditions. For **a** and **b**, lines show the mean of the rolling mean (centred window size = 3) and shaded ribbons the standard deviation of four replicates. In **d**, each line shows an independent competition experiment ($n = 2$ for each condition). Points on the lines show flow cytometry samples. In **e**, the lines show the rolling mean (centred window size = 3) of each independent experiment.

off), we return to a situation in which co-existence is not possible and one or the other strain will go extinct.

Our estimates of bacteriocin basal expression from Fig. 4d, suggest that stable co-existence is theoretically possible with a high dilution rate. However, the range of 3OC6-HSL production rates within which we can operate is very small, suggesting achieving this would be difficult. We ran co-cultures in a plate reader with increasing amounts of arabinose, at a starting density which, based on results in Fig. 3, we believe to produce dynamics akin to a dilution rate of 0.4 hr$^{-1}$ (Fig. 8a). Without arabinose, the engineered strain dominates the culture, as predicted. However, at 10 mM arabinose, we see co-existence for the length of the experiment, mimicking the dynamics seen in the simulation. It should be noted that the plate reader is a limited

approximation of the chemostat environment assumed in our mathematical model. Despite this, we have managed to capture the predicted behaviour; the co-culture enters stationary phase after ~5 h (Fig. 8b) while the population ratios appear to have stabilised after only 2.5 h.

In order to challenge the system further and to determine whether we could produce co-existence for a prolonged period, we repeatedly diluted and sampled co-cultures every 2 h (Fig. 8c). The volume that was removed and replaced with fresh media gave the equivalent chemostat dilution rate (see "Methods" section). The co-cultures were incubated in a plate reader and periodically diluted over an 18 h period. Samples were analysed using flow cytometry and population ratios were determined by clustering using fluorescence. The results (Fig. 8d) are qualitatively

comparable to the plate reader assay; without arabinose, the competitor is driven to extinction, with 10 mM arabinose, we see stable co-existence. At an intermediate concentration of arabinose, we can see the effects of changing the dilution rate, with the lower dilution rate resulting in competitor extinction as predicted by the model. Measurements of turbidity, while the plates were incubating in the plate reader, show the repeated dilution of the cultures and demonstrate that the cultures do not reach stationary phase (Fig. 8e). We see a shift in the equilibrium population ratios between the batch experiment (Fig. 8a) and the repeated dilution experiment (Fig. 8d). This is predicted from the modelling (Supplementary Fig. 6) as, in our batch experiments, the period that we use to approximate chemostat behaviour is 5 h. This results in a shift in the region of co-existence, leading to a higher equilibrium proportion of engineered strain for the same dilution rate and arabinose concentration.

**Identifying robust models for consortia control**. With the limitation that we only engineer a single strain in the consortia, there are other mechanisms that we could use instead of, or in addition to, the intercellular toxin used here: intracellular toxins have been used before for population control[29], and it has been suggested that expression of the immunity genes for toxins could be regulated to change the viability of strains[44]. The expression of all of these molecules could be constitutive or under the control of a quorum molecule also expressed by the engineered strain (Fig. 9a).

Using this set of possible parts, we can produce a model space that consists of 132 unique systems. We can assess the candidate models' ability to produce stable consortia using approximate Bayesian computation sequential Monte Carlo (ABC SMC) (Fig. 9b), which allows us to approximate model and parameter posterior probabilities by random sampling and weight assignment through a series of intermediate distributions[45–47]. The output of ABC SMC is an approximation of the posterior distribution of models and of the parameters for each model. The final model posterior distribution indicates which models have the highest probability of producing the objective behaviour—in this case coexisting communities at a steady state—while also accounting for system complexity (Occam's razor) (Fig. 9c). All of the top-performing systems use bacteriocin. Indeed, the best system without bacteriocins is ranked 75th. The best system overall requires the control of all four of the possible genes; the bacteriocin, immunity and antitoxin are repressed by the quorum molecule, while the intracellular toxin is induced by it (Fig. 9d,i).

The systems without intracellular toxin and antitoxin perform particularly well when one considers their relative simplicity. The best of these uses the quorum molecule to repress both the bacteriocin and immunity (Fig. 9d,ii). Intuitively, this leads to less killing of the competitor and increased susceptibility of the engineered strain at higher densities. A simpler system uses the quorum molecule only to repress the immunity gene (Fig. 9d,iii). The simplest system to robustly achieve stable consortia is the system that we built, requiring just repression of the bacteriocin by the quorum molecule (Fig. 9d,iv). The immunity gene is expressed in this system but, as its constitutive expression level does not need tuning, we consider this system simpler than Fig. 9d,iii.

None of the previously described systems[29,30,44] are able to produce stable communities. However, if we set our objective to co-existence, with both populations above a defined threshold, rather than stable communities, the other systems can achieve the goal (Supplementary Fig. 5). The theoretical system proposed by McCardell et al.[44] performs as well as the system we have constructed. Just as for stable communities, the addition of

quorum control over immunity expression, as in Fig. 9d,ii, improves robustness further. These results demonstrate that we have constructed a "best in class" system using only a limited number of parts, which is expected to be more robust than existing systems. It also provides a path to implementing more robust systems in the future.

**Discussion**

We have demonstrated that bacteriocins can be used to control the relative fitness of strains in competition with one another. The environment is a key determinant of strain fitness. We have shown that the dilution rate alone can be used to switch between the domination of a faster-growing strain and an amensal strain. Although plate reader assays, in which cultures grow to saturation on a limited substrate, do not fully reflect the chemostat environment which we have predominantly used for our mathematical modelling, computational simulations of a periodically diluted environment and our experimental results closely follow the model predictions. The inclusion of exogenous control through 3OC6-HSL allowed us to demonstrate the dose-dependent expression of a bacteriocin. However, we were only able to achieve a ten-fold change in expression, while the modelling results suggest an increase to 100-fold would enable a system with true switching ability regardless of the community composition. Using 3OC6-HSL also enabled us to extend the system to an autonomous one in which population density provided the switch from engineered strain to competitor strain domination. In our implementation arabinose had to be added to the environment to find the region of co-existence, though the same could be achieved with a library of different strength constitutive promoters without the need for a specific environment. The ability to alter quorum molecule production rate, through induction or repression, could be used in certain applications to respond to environmental change; for example, if a particular nutrient is present the rate could be changed to pre-empt a physiological change in the competitor.

The dynamic control of intracellular toxins through quorum sensing has been demonstrated before for community control[28,29]. However, such self-limiting systems are susceptible to loss-of-function mutations and require the engineered strain to grow faster than the competitor, or control of both strains. Although intercellular toxins have been used for synthetic ecologies[38], autonomous control has never before been demonstrated. Further, the embrace of competitive exclusion, although present, has not been explicit in any synthetic community control system; indeed it is necessary for our system. Uniquely, this allows the construction of synthetic communities while only requiring the engineering of a single strain. The theoretical upgrades that were suggested in Fig. 9 remove the requirement for competitive exclusion by the addition of self-limitation. We have previously discovered such motifs to improve the robustness of community control architectures[45].

If the leakiness of bacteriocin expression can be reduced, community stability at very low dilution rates (and therefore very low nutrient conditions) can be achieved. Further, as we have demonstrated that periodic rather than constant dilution of the environment can produce the desired dynamics, this suggests that this system may be able to function in "wild" environments in which resources are scarce and appear sporadically. In this work, we have demonstrated the system functioning for two strains. We believe this could be utilised for niche creation in native microbiomes; a single abundant resident strain could be targeted by our engineered strain rather than attempting to control all microbiome constituents. The development of biosensors for medical use requires confidence in the residence and, for quantitative

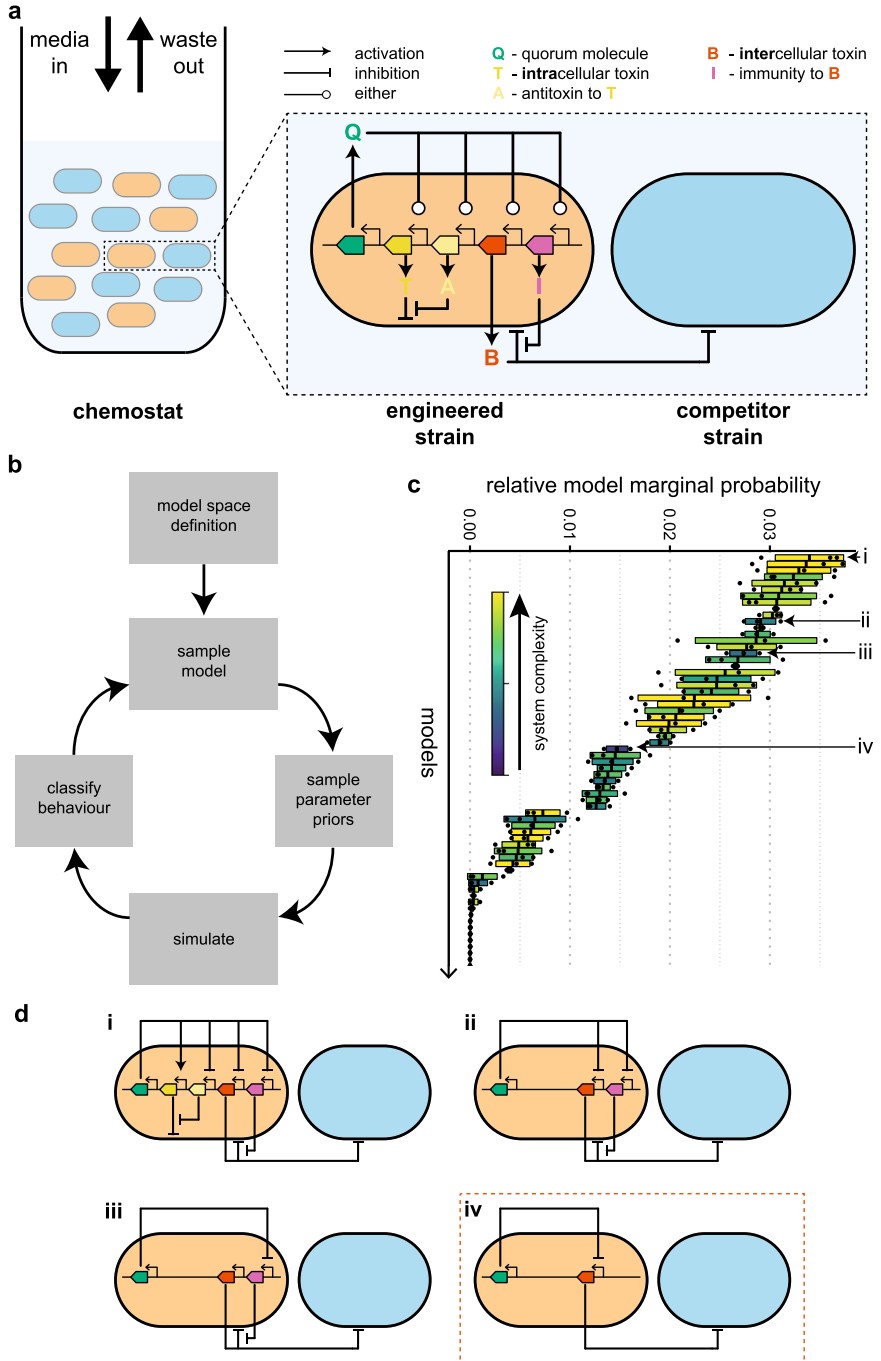

**Fig. 9 Exploring the space of possible models of population control to find the best system. a** Schematic of all possible components for single strain engineering. The engineered strain can produce an intercellular toxin and its antitoxin and an intracellular toxin and its antitoxin. All of these can be positively or negatively regulated by a self-produced quorum molecule, allowing density-dependent control of the engineered cell's actions. **b** Once the model space has been defined, we explore it using ABC SMC to determine which models with which parameters are capable of producing stable coexisting populations. **c** The ordered model marginal posterior probabilities. The long tail of extinct models has been trimmed. The boxes for each model are coloured according to the model's complexity; models with a larger number of parameters are closer to the yellow and fewer parameters are close to blue. **d** The best system (i) requires control over the expression of four genes. Systems that do not require intracellular toxin or immunity perform well for their level of complexity (ii–iv). In fact, the simplest system able to produce stable co-existence is the system that we have developed (iv). In **c**, each box is centred at the mean, with the left and right edges showing the standard deviation, of three replicates, each containing 18,000 particles. Black points show the outcome of each replicate.

rather than just qualitative sensors, the population size of the engineered strain at a target location. Delivery of therapeutics by engineered bacteria requires even more confidence in the dynamics of the engineered population in order to prevent over- or under-dosing[18].

We have engineered a proof-of-principle population control system using a monoculture as the native community in order to establish a niche. Future work will involve the implementation of the system in more complex communities[48,49] and in animal models including *C. elegans*[50]. In previous work, we have shown

mathematically that bacteriocins can be used to control more complex consortia[45]. Extensions of the system demonstrated here with additional bacteriocins and quorum molecules could allow its use in industrial biotechnology for the control of communities in bioreactors in which division-of-labour is desirable to prevent burden. Our theoretical model exploration has demonstrated that the system we have implemented is the simplest to produce stable co-existence of competing populations; performing better than the intracellular toxin approaches used to date. Further, it has suggested designs to upgrade the system in order to make the community control more robust. This method of model exploration could easily be extended to explore a model space in which the requirement for single strain engineering is dropped or other control mechanisms, such as substrate cross-feeding, are introduced.

The last decade has seen applied microbiology, particularly synthetic biology, recognise the importance of context, be it compositional, intracellular or environmental[51,52]. However, community context has largely been neglected, with the consequence that we have limited ourselves to building systems of single, homogeneous populations that are only capable of functioning in controlled environments. Contending with intracellular context has required the use of feedback to take into account the cell's response to our demands[9,53]. By embracing competitive exclusion, we have shown that feedback is also a crucial component in the construction of stable synthetic microbial communities and have demonstrated how to work with community context rather than against it.

## Methods

**Strains and plasmids**. A number of plasmids were created for this work, listed in Supplementary Table 3. Polymerase chain reaction primers used to create these plasmids are listed in Supplementary Table 5. All plasmid sequences were confirmed with Sanger sequencing and the plasmid maps are available online. Where the created plasmids have used SEVA vectors[54], the SEVA naming convention for antibiotic resistance and origin of replication has been adhered to, though the "SEVA" prefix has been removed as the plasmids do not necessarily adhere to the SEVA standards. The SEVA convention for the naming of the "cargo" region has not been adhered to as the extensibility of this naming convention is limited.

pBAD-mTagBFP2 was a gift from Vladislav Verkhusha (Addgene plasmid 34632; http://n2t.net/addgene:34632; RRID:Addgene_34632)[55]. pTD103luxI_sfGFP was a gift from Jeff Hasty (Addgene plasmid 48885; http://n2t.net/addgene:48885; RRID:Addgene_48885)[56]. pTNS1 was a gift from Herbert Schweizer (Addgene plasmid 64967; http://n2t.net/addgene:64967; RRID:Addgene_64967)[57].

The bacterial strains used in this work are detailed in Supplementary Table 4. Two strains were produced with chromosomal integration of Gentamicin resistance and fluorescence (CFP or mCherry) using the mini-Tn7 transposon system[58] with plasmids from the SEVA collection[54]. Electrocompetent cells were produced using the mannitol-glycerol step protocol[59].

**Plate reader competition assays**. Overnight cultures of the relevant strains (Competitor = EcM-Gm-CFP, Engineered strain = E. coli JW3910: p63_AF043 + pMPES_AF01) were grown, from single colonies picked from selective LB agar plates, in 5 mL of supplemented M9 media (0.4% glycerol, 0.2 % casamino acids) containing all necessary antibiotics. The overnight cultures were diluted 1:1000 into fresh M9 media containing 10 μg/mL Gentamicin and relevant inducers and grown for 6 h. The cultures were diluted to an OD (700 nm) of 0.1 using fresh M9 media containing 10 μg/mL Gentamicin and relevant inducers. The cultures were then mixed at the specified ratio. 10 μL of the culture was inoculated into 115 μL of media containing Gentamicin and relevant inducers in the wells of a 96 well microtitre plate and the plate was covered with a Breathe-Easy sealing membrane. The plate was placed in a plate reader (Tecan Spark) and grown for shown number of hours at 37 °C with continuous double orbital shaking (2 mm, 150 rpm). Measurements of absorbance (600 nm and 700 nm) and fluorescence (CFP: ex = 430 nm, em = 490 nm, GFP: ex = 485 nm, em = 530 nm, mCherry: ex = 575 nm, em = 620 nm) were taken every 20 min using Tecan Spark Control software. Data were processed using FlopR[40] to produce population and subpopulation curves, and plotted using R[60] with ggplot2[61] and dplyr[62].

**Deviations from the above**. Figure 1b: Wells in the microtitre plate were inoculated with 200 μL of culture and a magnetic, automatically removable, plastic lid was used to cover the plate. Measurements were taken every 45 min.

Figures 2 and 3: Rather than inoculating 115 μL of media in each well with 10 μL of culture, the first row of the plate was inoculated with the culture at 0.1 OD$_{700}$. Subsequent rows were serial dilutions with a dilution factor of 0.375 i.e., by the 8th row the culture was at ~1000th the density of the culture in the first row. GFP fluorescence was measured with emission = 535 nm and mCherry fluorescence was measured with excitation = 561 nm. Figure 5: A large amount of time taken to dilute culture to 0.1 OD$_{700}$ meant that by the time cultures were mixed, some cultures, notably the competitor cultures, were no longer at the desired OD. As such, initial ratios were not simply the mixing ratios. We instead determine the initial population ratios from the plate reader data. Figure 8d, e: The plate was covered with a magnetic, automatically removable, plastic lid to allow for regular dilutions. Every 2 h the plate was removed from the plate reader, a given volume was pipetted from each well into a flow cytometry sample plate (containing PBS for a total well volume of 125 μL), and the same volume of fresh media with the appropriate inducers and antibiotics was pipetted in. The transfer volume was calculated from the desired dilution rate using the exponential function $V_{trans} = V_{total} - V_{total}/\exp(D \times \delta t)$, where $\delta t$ is the period between dilution (2 h), $V_{total}$ is the total culture volume in each well (125 μL) and $D$ is the desired dilution rate. Flow cytometry data were analysed using FlopR and a custom R script, utilising flowCore[63], for population clustering (Supplementary Fig. 7).

**Agar plate spot inhibition assay**. Cultures of the bacteriocin producing strain (E. coli MG1655: p23_AF041 + pMPES_AF01) were grown overnight in supplemented M9 media without antibiotics but with the recorded concentration of inducer. A culture of a bacteriocin sensitive strain (EcM-Gm-CFP) was also grown overnight. A one-well plate was filled with 30 mL of 1% LB agar with Gentamicin and allowed to set. 10 mL of 0.5% LB agar with Gentamicin was inoculated with 100 μL of the overnight culture of the bacteriocin sensitive strain, poured over the one-well plate and allowed to set for 1 h. The overnight cultures for sampling (induced bacteriocin producing strain and non-producing control) were spun down at 2500 × g for 10 min and 3 μL of supernatant from each culture was spotted on to the surface of the lawn and the plate was allowed to dry for a further hour—the Gentamicin in the agar prevents any remaining cells in the supernatant from growing and negates the need to filter, and therefore potentially lose bacteriocin. The plate was placed in an incubator at 37 °C for 20 h. The plate was visualised in a UVP GelDoc-it imager using the epi-white light source.

An image processing pipeline was used to extract quantitative inhibition zones from the image, Supplementary Fig. 3. The image was flattened using a gradient mask in Adobe Photoshop (CC 2019) to remove illumination differences between the centre and edges of the plate. Then, in Fiji[64], the image was thresholded - using Auto Threshold with Method = default - to convert from greyscale to black and white. A mutational close was performed - Iterations = 100, Count = 5 - to remove graininess from the image. A Gaussian blur filter was then applied - sigma = 10.00 - to remove blank areas at the centre of inhibition zones left by the pipette tip. Finally, a threshold was applied, with the same settings as previously, and the pixels within each inhibition zone counted using the Fiji Analyze Particles function. A Hill function was fitted to the data using RStan[65].

**Characterisation of genetic circuits**. All plasmids were transformed into E. coli MG1655 by electroporation for characterisation. Overnight cultures of the relevant strains were grown in supplemented M9 media with the appropriate antibiotics in a shaking incubator at 37 °C and 200 rpm. After 16 h of growth, the cultures were diluted 1:1000 into fresh supplemented M9 media with antibiotics and grown for 6 h in a shaking incubator at 37 °C and 200 rpm. After 6 h the cultures were diluted 1:100 into fresh supplemented M9 media with antibiotics and induced with the relevant concentration of inducer. 200 μL of each induced culture was then pipetted into a clean 96 well microtitre plate and sealed with a Breathe-Easy sealing membrane. It was incubated at 37 °C and had constant double orbital shaking for 16 h.

The microtitre plate was then removed and 1 μL of culture from each well was used to inoculate 200 μL of PBS in a clean round-bottom 96 well microtitre plate. Flow cytometry was performed on an Attune NxT Acoustic Focusing Cytometer with Attune NxT Autosampler (ThermoFisher Scientific, UK). The Attune NxT Autosampler was set to sample 20 μL from each well with 2 mixing cycles and 4 rinses between each sample. Forward and side scatter height and area measurements were always recorded. Height and area measurements in the appropriate fluorescent channels were also recorded. The fluorescence channels are detailed in Supplementary Table 6. The flow cytometry data was processed using FlopR[40] to remove debris and doublets, and Hill functions were fitted using RStan[65].

**Reporting summary**. Further information on research design is available in the Nature Research Reporting Summary linked to this article.

## Data availability

All data, including plasmid maps, are available at https://doi.org/10.5281/zenodo.4560355. Any other relevant data are available from the authors upon reasonable request.

## Code availability

The model exploration using ABC SMC[45] is available at https://github.com/ucl-cssb/AutoCD. Plate reader and flow cytometry data was processed using FlopR v0.3.0[40], which is available at https://github.com/ucl-cssb/flopr. A Python notebook for simulation of the chemostat model and an R script for plotting the plate reader data are included in the data repository https://doi.org/10.5281/zenodo.4560355.

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

## Acknowledgements

We thank Y. Kaznessis and K. Geldart for plasmid pMPES:V, V. de Lorenzo and E. Martinez-Garcia for all SEVA plasmids, and J. Ward for the *E. coli* MG1655 strain. We also thank N. Collant for guidance during some experiments. A.J.H.F. and C.P.B. received funding from the European Research Council (ERC) under the European Union's Horizon 2020 research and innovation programme (Grant No. 770835). C.P.B. received funding from the Wellcome Trust (209409/Z/17/Z). A.J.H.F. also received funding from an EPSRC studentship through the CoMPLEX Centre for Doctoral Training (Award Ref. 1327857). B.D.K. was funded through the BBSRC LIDo Doctoral Training Partnership.

## Author contributions

Conceptualisation, A.J.H.F. and C.P.B.; formal analysis, A.J.H.F. and B.D.K.; investigation, A.J.H.F. and B.D.K.; methodology, A.J.H.F., B.D.K., M.S. and C.P.B.; project administration, C.P.B.; software, A.J.H.F. and B.D.K.; supervision, M.S. and C.P.B.; writing - original draft, A.J.H.F. and B.D.K.; writing - review and editing, A.J.H.F., B.D.K., M.S. and C.P.B.

## Competing interests

The authors declare no competing interests.
