## [Peer Review File · Nature Communications]

Reviewers' Comments:

Reviewer #1:

Remarks to the Author:

In this study, the authors investigate an interaction between two strains of *E. coli*, where one of them is engineered to up- and downregulate bacteriocin production in response to certain stimuli, and the engineered system allows for co-existence of these two strains over several hours. The authors also describe mathematical modelling to investigate different versions of this engineered "community control" system.

I find the system they describe to be intriguing and the manuscript well-written overall. My only "major" complaints (really not that major) are the following:

a) The definition of what constitutes a "stable" community.

It appears that the authors use the word "stabilized" to describe a community where competitive exclusion does not happen anymore, i.e. a community that allows for the co-existence of the two focal strains. In ecological theory, however, stability is most often used in the context of resilience, i.e. communities that return to an equilibrium after significant perturbation (e.g. reaching equilibrium, then near-complete removal of one strain, and then return to equilibrium). The "stability" shown in e.g. Fig. 7C appears to only show one "cycle" where competitive exclusion kicks in, the engineered strain makes a bit of bacteriocin and then reaches a balance with the competitor. But what is lacking here is the crucial step of disrupting the system and see if the community can "bounce back" to equilibrium, to fulfil the most commonly used definition of stability. Given how the system is set up I actually expect these simple communities to be formally stable, but I would like to point out that this was technically not shown in this study, so the authors might want to reconsider some of the semantics here.

b) The abstract stating that the study considers "multi-species"(L12) and "two-species consortia" (L16).

The presented experiments cover a pairwise interaction between two strains of the same species (*E. coli*), so the abstract should be rephrased to reflect this.

c) The limitations of using bacteriocins.

Most bacteriocins are extremely narrow in their target range and can only target strains within the same species, or sometimes closely related species. The use of a bacteriocin (microcin V) in this system thus severely limits its potential for the (discussed) purpose of controlling natural microbiota. Realistically, you can probably only control one strain at a time (or maybe a really closely related species like *Salmonella*, if we are being generous). So, I would argue that the system described here - using an engineered strain producing a single, narrow-range bacteriocin - is not as broadly applicable as the authors make it out to be, and it would not allow control of "a community through a single constituent" (L47) if we consider naturally occurring communities (e.g. the microbiome) which are made up of hundreds of species. In L206, the authors state that "this allows the construction of synthetic communities while only requiring the engineering of a single strain, a feature that enables applications in which an engineered population can coexist with native microbiomes.", except the single engineered strain only kills a very specific type of competitor (the one with the specific bacteriocin receptor on its membrane), so application is limited to "communities" of *E. coli* and relatives, and not to actual mixed-species native microbiomes. I suggest the authors discuss this caveat more explicitly in the introduction and discussion.

d) Describing the system as "fully autonomous".

The authors postulate that their system does not need "environmental control" to achieve stability, but I would say adding an external stimulant such as a high concentration of arabinose (as shown in Fig. 7C) is environmental control. It basically means that parts of the system need to be cranked up to full power (maximally induced production of LuxI, which regulates the production of

QS molecules) by an externally added chemical stimulus to achieve strain co-existence. I would argue that the "true" benchmark of an autonomously self-regulated system is one that can be perturbed (see comment on "stability" above) and will return to a pre-defined stable state without external addition of any direct stimulant of the system.

In summary, I think this study covers an interesting topic, the experiments are well described in the methods and the analyses are appropriate. I think the introduction and discussion could benefit from more fully addressing the obvious limitations of this system for actual multi-species community control, and the use of labels like "stability" and "full autonomy" should be considered more carefully throughout the manuscript.

Minor comments:

1. L67 (whole paragraph); what is the rationale for the starting ratios of 75:25 here? Competition dynamics in this system are highly frequency-dependent (as the authors show nicely in Fig. 2C), so it would be helpful if starting ratios were always explicitly mentioned when describing competition outcomes in the results section.
2. L73-74; I am a bit confused about this statement. How does this follow from the results in Fig. 2C? Elaborating on this could help the reader follow the logic a bit better.
3. L124 "use population density for control"; adding arabinose forces them to produce large amounts of quorum molecules per cell, so how does this interfere with them actually sensing density? Does this simply lower the QS threshold, so that they switch off bacteriocin production faster/earlier during the competition? The interaction between arabinose induction and sensing density could be explained a bit more in detail here.
4. L158 typo; the viability [of] strains?
5. L196 "first time ... dose-dependent expression of bacteriocins". Colicins are routinely expressed in a dose-dependent way by addition of mitomycin C or other DNA-damaging agents (e.g. <https://journals.plos.org/plosone/article?id=10.1371/journal.pone.0119124>). Perhaps this statement should be restricted to microcins if dose-dependent induction was indeed never shown before for them.
6. L269; shouldn't this be strain JW3910? MG1655 is the fast grower non-producer I thought?
7. L275 typo "minutes"
8. Figures – what is the difference between the experiments shown in Fig. 1B and those shown in Fig. 2? Adding description of experiments to Fig.1 legend would help here.
9. Fig. 2C, the y axis is a bit confusing here – are those discrete fractions? So we are looking at four experiments? But the colour blocks don't correspond (or line up well to) the y axis ticks?

Reviewer #2:

Remarks to the Author:

The biosynthesis of compounds often requires engineering and functioning of complex metabolic pathways in a single population, which often leads to heavy metabolic load. Division-of-labor can minimize the burden, but competitive exclusion inevitably results in the removal of less fit members over time. In this manuscript, the authors propose a strategy to stabilize multi-species communities by leveraging a toxin producing system. They showed experimentally and mathematically that such a system can produce stable populations with a composition that is tunable by controllable parameters.

Although composition control is an important topic for microbial ecosystems, the advantages and utilities of the proposed approach are not convincing due to the following three reasons.

First, in my view, cross-feeding is the simplest and probably the most robust strategy for engineering stable ecosystems. Such an approach indeed has been demonstrated by previous studies. I doubt that the method in the paper outperforms cross-feeding.

Second, this controlling achieves population structure control by killing fast-growing cells, which is a constant, heavy waste of energy and resources.

Third, the authors also argue that engineering multiple species of an ecosystem is a disadvantage of current synthetic ecosystems. However, the need to engineer multiple species does not imply a higher degree of complexity or engineering challenge. In fact, targeting on multiple species within a community offer a broader spectrum of tunability and programmability compared to the engineering of a single strain.

Other comments:

1) The authors argued that the single-strain based population control offers many applications, but the paper does not provide examples. I think having a demonstration of the utility of the controlling method is highly desirable.

2) In the study, toxin is expressed from plasmids without the supplement of antibiotic in the co-culture. How stable the plasmid is during long-term culturing?

3) The authors cultured the consortium using a microplate reader but modeled the ecosystem with chemostat settings, which led to inconsistency as they pointed out. I suggest the authors to either revise the models to include nutrient as a variable or to conduct experiment in chemostat. In particular, conducting chemostat experiment will be highly valuable for validating the long-term stabilizing ability of the ecosystem without nutrient depletion.

Reviewer #3:

Remarks to the Author:

Summary of Results

This paper focuses on the development of an engineered *E. coli* strain that can regulate the composition of a two-strain *E. coli* consortia, in which the second strain is not engineered. The authors harnessed the killing property of bacteriocins to control the population growth of the non-engineered fast-growing strain, which they call the competitor strain. The authors built a mathematical model to simulate the growth of the two-strain consortia in a chemostat and study the effect of the chemostat dilution rate on the rate of killing of the competitor strain. They confirmed via plate-reader experiments that the dilution rate can be used to favour either strains given their current population density. The authors recognised that tuning the dilution rate may not always be possible given the context in which the consortium is grown. As a result they implemented a circuit inducible by C6-HSL to control the production rate of the bacteriocin. Results show that C6-HSL can control the production of bacteriocin, and hence it can control the concentration of the competitor strain. Next they built a circuit which autonomously produces C6-HSL to control the killing rate of the competitor strain. They characterised each module in the circuit: the module expressing LuxI under the control of arabinose, the module expressing TetR under the control of C6-HSL and the module expressing the bacteriocin under the control of TetR. Finally, they showed that the concentration of arabinose injected into the system provides a way to screen how much C6-HSL needs to be produced by the engineered strain in order to maintain the engineered strain and the competitor strain at a stable ratio in the consortium.

Major Comment

Further experiments should be carried out to determine whether the ratio achieved by the circuit is indeed stable. Currently the experimental validations of the chemostat simulations have been done in microplates. This showed that the ratio between the competitor and engineered strains become stable as the strains transition to stationary phase, when cells are not expected to grow anymore

and hence the effect on the population ratio would be minimal. Without an experiment presenting results maintaining cell consortia ratios in exponential phase as in the presented simulations, there is not enough experimental evidence that the proposed ratiometric control architecture works adequately in vivo. More evidence is thus needed to confirm that such stable ratio could be obtained and maintained in exponential growth. Without such additional evidence, it is difficult to ascertain that the population control system proposed by the authors is indeed capable of maintaining a specific ratio.

Specific Comments

1. ****Page 1, line 18 - Introduction:****

Replace "has enabled" by "have enabled".

2. ****Page 1, line 19 - Introduction:****

The authors may want to clarify what "enabling the construction of chromosome scale, synthetic biological system" means in this context. Is the sentence referring to the current capability of engineering genomes, or synthesising large pieces of DNA, or constructing large constructs? Since the paper does not present a circuit that is integrated into the genome, this sentence can spread confusion unless you have plans of integrating your circuit into the genome, which should perhaps be mentioned later.

3. ****Page 2, line 60-61 - Results - Bacteriocins enable population control at a distance:****

Could the authors provide more explanation as to what "strong" and "weak" means in this context? Would "fast growing" and "slow growing" be more suitable in this context?

Could the authors justify their choice of using the JW3910 strain? Would it be relevant to mention that the JW3910 strains is a metB knockout? Given that the strain is grown in M9 media supplemented with casaminoacids, did the use of JW3910 decrease the growth rate compared to the wild-type Keio strain?

4. ****Page 2, line 62-63 - Results - Bacteriocins enable population control at a distance:****

The authors may want to rephrase this sentence to make it more clear that they are investigating a way to reduce the growth rate of the engineered strain by expressing mCherry in the cell. Given that mCherry is expressed from the genome, did the authors notice a decrease in growth rate compared to the strain that is not expressing mCherry? Could these data be provided as supplementary information?

4. ****Page 3, line 84-85/Figure 3B - Results - Restricting growth favours bacteriocin producers over fast growers:****

Could the authors provide more explanation as to why serially diluting cultures emulates growth in a chemostat? Intuitively doing serial dilution will change the consortium initial density, as shown in Figure 3B. However that would imply that the cells starting from a smaller density should remain in lagphase for longer than consortia starting with a higher density. Figure 3B does not show this extended lagphase for low starting densities. Could the authors explain why that might be the case?

6. ****Page 5, line 145-146 - Results - Fully autonomous community regulation:****

Could the authors provide more detail as to why they are working with a degradation tag on the C6-HSL synthase (LuxI-LAA)? Could removing the tag improve the production rate of C6-HSL as

the synthase would not be degrading so quickly anymore?

7. **Page 7, line 184 - Results - Identifying robust models for consortia control:**

The McCardell paper presents an architecture to achieve co-existence, but has not demonstrated experimentally that they could achieve co-existence of their bacterial strains. Do the authors refer to the architecture or the results of this paper?

8. **Page 7, line 193-196 - Discussion:**

The authors claim that their batch experiments carried in microplates follow the results of their in-silico simulations. However, the stable ratio that they observe experimentally seem to happen during the stationary phase during which cells do not divide anymore, hence leading to a stabilised ratio which does not demonstrate that the same would happen in a continuous culture. Without an experiment presenting results maintaining cell consortia ratios in exponential phase as in the presented simulations, there is not enough experimental evidence that the proposed ratiometric control architecture works in vivo.

9. **Page 7, line 205-206 - Discussion:**

One could argue that all previous systems aiming to engineer co-cultures have exploited the principle of competitive exclusion as they all, in their own way (e.g. syntrophic co-cultures), created systems that push strains to rely on the existence of the other to survive in order to modulate their respective growth rates in the co-culture. The authors might want to discuss this a bit more.

10. **Page 7, line 206-210 - Discussion:**

The authors should highlight that the growth context for these two types of applications (engineering the microbiome and production in bioreactors) are very different and require different circuit properties. Have the authors explored (through simulations) whether their circuit has the potential to work in stationary phase, as would be the case in the context of the gut?

11. **Page 8, line 228 - Methods - Strains and Plasmids:**

Neither the plasmid maps nor the plasmid sequences have been added to the supplementary file provided. Will they be provided in a separate file later on?

12. **Page 9, line 260 - Methods - Deviations from the above:**

There seems to be a typo with the OD reported. It currently says "01. OD_{700}". This should probably be either 0.1 OD or 0.01 OD.

13. **Page 10, line 274-275 - Methods - Agar Plate Spot Inhibition Assay:**

The authors should consider describing which culture (engineered or competitor) they are referring to when they write "the overnight cultures for sampling".

14. **Figure 1B:**

Please provide a legend for the black curve? (E+C)

Please mention explicitly in the legend that the data were obtained experimentally (and not by simulation). Are the data provided the mean of biological replicates? If so, please include error bars. If biological replicates were too noisy, please show the replicates data in the supplementary

material.

15. **Figure 2:**

Were biological replicates performed for both panels A and B? If so, please indicate this in the legend.

16. **Figure 4C:**

Could the authors confirm that the image of the agar plate spot inhibition assay has been inverted when imaging with the GelDoc? It is currently counter-intuitive that the inhibition zone is dark while we would expect to see a "hole" (so lighter) in the competitor strain lawn.

17. **Figure 5:**

Please provide information about the biological replicates in the legend of Figure 5 as well.

18. **Figure 7C:**

Please display the OD for each of the experimental graphs. Can you claim that the co-culture is stable if it is in stationary phase? I recommend the authors provide a complementary experiment that keeps the cells in exponential phase over time. (if a chemostat chamber isn't available to the authors, an experiment performed by manually diluting the cultures before they reach stationary phase so as to keep the cells in exponential growth would provide information about the ability of the proposed ratiometric control system to work as intended)

REVIEWER COMMENTS

Reviewer #1 (Remarks to the Author):

In this study, the authors investigate an interaction between two strains of *E. coli*, where one of them is engineered to up- and downregulate bacteriocin production in response to certain stimuli, and the engineered system allows for co-existence of these two strains over several hours. The authors also describe mathematical modelling to investigate different versions of this engineered "community control" system.

I find the system they describe to be intriguing and the manuscript well-written overall. My only "major" complaints (really not that major) are the following:

a) The definition of what constitutes a "stable" community.

It appears that the authors use the word "stabilized" to describe a community where competitive exclusion does not happen anymore, i.e. a community that allows for the co-existence of the two focal strains. In ecological theory, however, stability is most often used in the context of resilience, i.e. communities that return to an equilibrium after significant perturbation (e.g. reaching equilibrium, then near-complete removal of one strain, and then return to equilibrium). The "stability" shown in e.g. Fig. 7C appears to only show one "cycle" where competitive exclusion kicks in, the engineered strain makes a bit of bacteriocin and then reaches a balance with the competitor. But what is lacking here is the crucial step of disrupting the system and see if the community can "bounce back" to equilibrium, to fulfil the most commonly used definition of stability. Given how the system is set up I actually expect these simple communities to be formally stable, but I would like to point out that this was technically not shown in this study, so the authors might want to reconsider some of the semantics here.

We use the term "stable" here to discriminate between co-existence and stable population equilibria, the latter being what we aim to achieve,. We have included extra experimental data in, what is now, Figure 8 in which the co-cultures are repeatedly diluted every 2 hours for 18 hours. Although this is intended to approximate chemostat dilution, it provides a repeated periodic disruption to the community, as suggested by the reviewer, and confirms the stability of the co-cultures.

b) The abstract stating that the study considers "multi-species"(L12) and "two-species consortia" (L16).

The presented experiments cover a pairwise interaction between two strains of the same species (*E. coli*), so the abstract should be rephrased to reflect this.

We have changed the abstract to reflect that a two-strain community was used.

c) The limitations of using bacteriocins.

Most bacteriocins are extremely narrow in their target range and can only target strains within the same species, or sometimes closely related species. The use of a bacteriocin (microcin V) in this system thus severely limits its potential for the (discussed) purpose of controlling natural microbiota. Realistically, you can probably only control one strain at a time (or maybe a really closely related species like *Salmonella*, if we are being generous). So, I would argue that the system described here - using an engineered strain producing a single, narrow-range bacteriocin - is not as broadly applicable as the authors make it out to be, and it would not allow control of "a community through a single

constituent" (L47) if we consider naturally occurring communities (e.g. the microbiome) which are made up of hundreds of species. In L206, the authors state that "this allows the construction of synthetic communities while only requiring the engineering of a single strain, a feature that enables applications in which an engineered population can coexist with native microbiomes.", except the single engineered strain only kills a very specific type of competitor (the one with the specific bacteriocin receptor on its membrane), so application is limited to "communities" of E. coli and relatives, and not to actual mixed-species native microbiomes. I suggest the authors discuss this caveat more explicitly in the introduction and discussion.

We have added further context to the introduction regarding the spectrum of bacteriocin activity. We have highlighted that, although most bacteriocins seem to show a narrow spectrum of activity, some do exist with a broader spectrum. We also highlight that there have been successful attempts to engineer bacteriocins to change their targets and increase their spectrum and that many species express multiple bacteriocins.

We agree that the system demonstrated here is limited in its ability to broadly act against a microbiome. However, coexistence may be achieved by targeting a single, abundant, strain in the microbiome for niche creation. To this effect, the bacteriocin that we used can be exchanged for another, or indeed combined with others, in order to target another niche.

d) Describing the system as "fully autonomous".

The authors postulate that their system does not need "environmental control" to achieve stability, but I would say adding an external stimulant such as a high concentration of arabinose (as shown in Fig. 7C) is environmental control. It basically means that parts of the system need to be cranked up to full power (maximally induced production of LuxI, which regulates the production of QS molecules) by an externally added chemical stimulus to achieve strain co-existence. I would argue that the "true" benchmark of an autonomously self-regulated system is one that can be perturbed (see comment on "stability" above) and will return to a pre-defined stable state without external addition of any direct stimulant of the system.

The use of arabinose for induction of LuxI enabled us to find a region of LuxI expression which allows coexistence. This could have been achieved with constitutive promoters but would have involved extra cloning for every expression level modification. Further, although the environment is established to illicit a particular response, there is no external interaction with the environment other than the setup. As such, we believe it is valid to describe the system as fully autonomous. We have added some clarification to the discussion.

In summary, I think this study covers an interesting topic, the experiments are well described in the methods and the analyses are appropriate. I think the introduction and discussion could benefit from more fully addressing the obvious limitations of this system for actual multi-species community control, and the use of labels like "stability" and "full autonomy" should be considered more carefully throughout the manuscript.

Minor comments:

1. L67 (whole paragraph); what is the rationale for the starting ratios of 75:25 here? Competition dynamics in this system are highly frequency-dependent (as the authors show nicely in Fig. 2C), so it would be helpful if starting ratios were always explicitly mentioned when describing competition outcomes in the results section.

We have added to the text that this particular timeseries was chosen as it most clearly showed competitive exclusion followed by bacteriocin killing. The timeseries was chosen *post hoc* from the data shown in Supplementary Figure 1. In principle several other timeseries could have been used but the period of competitive exclusion would be compressed and more difficult to see.

2. L73-74; I am a bit confused about this statement. How does this follow from the results in Fig. 2C? Elaborating on this could help the reader follow the logic a bit better.

We have removed this statement as it was a pre-emptive rewording of the following section.

3. L124 “use population density for control”; adding arabinose forces them to produce large amounts of quorum molecules per cell, so how does this interfere with them actually sensing density? Does this simply lower the QS threshold, so that they switch off bacteriocin production faster/earlier during the competition? The interaction between arabinose induction and sensing density could be explained a bit more in detail here.

It is correct that increasing arabinose induction, increases the production quorum molecules per cell and therefore reduces the population threshold at which the bacteriocin turns off. We have extended the explanation as suggested.

4. L158 typo; the viability [of] strains?

Corrected.

5. L196 “first time ... dose-dependent expression of bacteriocins”. Colicins are routinely expressed in a dose-dependent way by addition of mitomycin C or other DNA-damaging agents (e.g. <https://journals.plos.org/plosone/article?id=10.1371/journal.pone.0119124>). Perhaps this statement should be restricted to microcins if dose-dependent induction was indeed never shown before for them.

We thank the reviewer for bringing this literature to our attention. We have removed the “first time” from our statement.

6. L269; shouldn't this be strain JW3910? MG1655 is the fast grower non-producer I thought?

This assay was performed in MG1655 transformed with the relevant plasmids, before we decided to switch to JW3910 as the host strain. As this assay does not rely on relative growth dynamics, we believe it is reasonable to assume that the results can be extrapolated to the JW3910. The change of host strains was due to the modelling demonstrating that growth rate differences between engineered and competitor strains could be used to tune competition (which we could theoretically do using the auxotrophy), however this was not experimentally explored.

7. L275 typo “minutes”

Corrected.

8. Figures – what is the difference between the experiments shown in Fig. 1B and those shown in Fig. 2? Adding description of experiments to Fig.1 legend would help here.

Experimental differences are highlighted in the “Deviations from the above” subsection in the “plate reader competition assays” section of the Methods. Specifically, Figure 1B was performed with a slightly higher volume, a different plate sealing technology and a different measurement frequency, than Figure 2.

9. Fig. 2C, the y axis is a bit confusing here – are those discrete fractions? So we are looking at four experiments? But the colour blocks don't correspond (or line up well to) the y axis ticks?

Each horizontal represents a single timecourse, so there are four timecourses as the reviewer notes. The blocks are horizontally centred on the initial co-culture ratio. For clarity, we have changed the axis ticks to reflect these initial ratios.

Reviewer #2 (Remarks to the Author):

The biosynthesis of compounds often requires engineering and functioning of complex metabolic pathways in a single population, which often leads to heavy metabolic load. Division-of-labor can minimize the burden, but competitive exclusion inevitably results in the removal of less fit members over time. In this manuscript, the authors propose a strategy to stabilize multi-species communities by leveraging a toxin producing system. They showed experimentally and mathematically that such a system can produce stable populations with a composition that is tunable by controllable parameters.

Although composition control is an important topic for microbial ecosystems, the advantages and utilities of the proposed approach are not convincing due to the following three reasons.

First, in my view, cross-feeding is the simplest and probably the most robust strategy for engineering stable ecosystems. Such an approach indeed has been demonstrated by previous studies. I doubt that the method in the paper outperforms cross-feeding.

We agree with the reviewer that cross-feeding is a commonly used method for ecosystem engineering. Its prevalence in nature suggests that it is a robust way of achieving co-existence through mutual dependence. Indeed, we have recently demonstrated computationally that mutual dependence networks, in this case using bacteriocin repression, are the most robust at producing co-existence (Karkaria et al., *bioRxiv* 2020, accepted *Nat. Comms.*). However, such systems, whether using cross-feeding or any other mechanism for mutualism, require choice of compatible strains or engineering of all strains. Our system explicitly creates coexistence without that restriction, enabling the potential manipulation of extant ecosystems, whether they be wild microbiomes or synthetic microbiomes from pre-engineered strains. In this manuscript we have only made a comparison with systems that meet this requirement and make no claims in comparison with a cross-feeding system.

Second, this controlling achieves population structure control by killing fast-growing cells, which is a constant, heavy waste of energy and resources.

We agree that killing cells uses up resources but in cases where we wish to engineer niche creation, there may be no other choice. Indeed, bacteriocins have evolved for this precise reason; bacterial competition. Granted, in certain circumstances, there may be more energy efficient means to create stable ecosystems (though using cross-feeding may also be inefficient in particular cases).

We base our system on the assumption and observation that our engineered strain, and engineered strains in general, is burdened in comparison to native strains. It is the growth rate difference caused by this burden that we rely on to produce the competitive exclusion which counters the bacteriocin killing. Our modelling suggests that the system should function at very low dilution rates, and therefore very low resource input rates, as long as the leakiness of the bacteriocin expression is very low. Further, the bacteriocin kills cells by pore formation which will lead to the return of some resources to the environment. The upgrades suggested by the model exploration in Figure 9 (previously Figure 8) remove the need for the competitor to grow faster.

Third, the authors also argue that engineering multiple species of an ecosystem is a disadvantage of current synthetic ecosystems. However, the need to engineer multiple species does not imply a higher degree of complexity or engineering challenge. In fact, targeting on multiple species within a

community offer a broader spectrum of tunability and programmability compared to the engineering of a single strain.

We agree that more points of control in a system, by engineering more constituent members of the community, may allow for improvements in controllability. However, this extra engineering necessarily adds extra complexity, either in the design of feedback systems or the fine tuning of parameters. In a recent paper (Karkaria et al., biorXiv 2020, accepted Nat. Comms.), we demonstrated the diminishing returns of adding extra elements to a network for control of communities. In addition, our approach enables the construction of hybrid ecosystems, of wild or pre-engineered strains, rather than those which are entirely synthetic and designed from scratch for each application.

Other comments:

1) The authors argued that the single-strain based population control offers many applications, but the paper does not provide examples. I think having a demonstration of the utility of the controlling method is highly desirable.

We have added further context in the introduction and discussion. We believe, in its current form our system is best applied as a niche creation device in wild microbiomes. By targeting an abundant, resident species, our engineered strain could ensure its persistence with minimal disruption to the native microbiome. This is a crucial ability for engineered biotherapeutics in order to avoid potential problems with over- or under-dosing (Ozdemir et al., Cell Systems, 2018). As an example, butyrate production by probiotic bacteria has been demonstrated to be beneficial for gut health (Geirnaert et al., Sci. Rep., 2017), but over production of butyrate may be harmful (Hamer et al., Aliment. Pharmacol. Ther., 2007). Our system could be used to maintain a consistent level of butyrate producing bacteria, allowing butyrate to be sustained within a healthy window. Another example is the use of biosensing strains to detect and report on disease states *in vivo* (Danino et al., 2015). Without knowledge of the size of the reporting population, it is impossible to distinguish between a weak signal from a large population and a strong signal from a small population. Our system could enable more quantitative reporting from *in vivo* biosensors.

2) In the study, toxin is expressed from plasmids without the supplement of antibiotic in the co-culture. How stable the plasmid is during long-term culturing?

In previous work (Fedorec et al., iScience, 2019) we demonstrated the use of microcin-V as a plasmid stability mechanism, with effectiveness for more than 20 days of daily passaged culturing. The second plasmid, carrying the quorum signalling components, incorporates a Gentamicin resistance gene which we select for with Gentamicin in our growth media.

3) The authors cultured the consortium using a microplate reader but modeled the ecosystem with chemostat settings, which led to inconsistency as they pointed out. I suggest the authors to either revise the models to include nutrient as a variable or to conduct experiment in chemostat. In particular, conducting chemostat experiment will be highly valuable for validating the long-term stabilizing ability of the ecosystem without nutrient depletion.

The chemostat environment for the mathematical modelling was chosen so that steady state analyses could be performed. A growth substrate is included in this model and is important in the Monod growth kinetics in chemostat as well as in batch. We have included a new Supplementary Figure 6 comparing the community compositions achieved by chemostat and repeated-dilution

batch models. This comparison demonstrates that repeated batch experiments, with a dilution frequency of 2 hours or less, closely correspond to the chemostat predictions. We have included further experimental data in Figure 8 showing stable co-existence with repeated dilution, every 2 hours, for a duration of 18 hours.

Reviewer #3 (Remarks to the Author):

Summary of Results

This paper focuses on the development of an engineered *E. coli* strain that can regulate the composition of a two-strain *E. coli* consortia, in which the second strain is not engineered. The authors harnessed the killing property of bacteriocins to control the population growth of the non-engineered fast-growing strain, which they call the competitor strain. The authors built a mathematical model to simulate the growth of the two-strain consortia in a chemostat and study the effect of the chemostat dilution rate on the rate of killing of the competitor strain. They confirmed via plate-reader experiments that the dilution rate can be used to favour either strains given their current population density. The authors recognised that tuning the dilution rate may not always be possible given the context in which the consortium is grown. As a result they implemented a circuit inducible by C6-HSL to control the production rate of the bacteriocin. Results show that C6-HSL can control the production of bacteriocin, and hence it can control the concentration of the competitor strain. Next they built a circuit which autonomously produces C6-HSL to control the killing rate of the competitor strain. They characterised each module in the circuit: the module expressing LuxI under the control of arabinose, the module expressing TetR under the control of C6-HSL and the module expressing the bacteriocin under the control of TetR. Finally, they showed that the concentration of arabinose injected into the system provides a way to screen how much C6-HSL needs to be produced by the engineered strain in order to maintain the engineered strain and the competitor strain at a stable ratio in the consortium.

Major Comment

Further experiments should be carried out to determine whether the ratio achieved by the circuit is indeed stable. Currently the experimental validations of the chemostat simulations have been done in microplates. This showed that the ratio between the competitor and engineered strains become stable as the strains transition to stationary phase, when cells are not expected to grow anymore and hence the effect on the population ratio would be minimal. Without an experiment presenting results maintaining cell consortia ratios in exponential phase as in the presented simulations, there is not enough experimental evidence that the proposed ratiometric control architecture works adequately *in vivo*. More evidence is thus needed to confirm that such stable ratio could be obtained and maintained in exponential growth. Without such additional evidence, it is difficult to ascertain that the population control system proposed by the authors is indeed capable of maintaining a specific ratio.

We agree with the reviewer that, once the co-culture reaches stationary phase, we cannot claim to have control over the stability of our co-culture. We have included the growth curves for the co-cultures in Figure 8 (previously Figure 7C). The growth curves show that stationary phase is reached after 5 hours but the community stabilises around 2.5 hours.

Further, we have included extra experimental data in, what is now, Figure 8 in which the co-cultures are repeatedly diluted every 2 hours for 18 hours. This demonstrates the stabilisation of co-cultures with an approximation of a chemostat environment, maintaining the cultures in exponential growth. A new Supplementary Figure 6 shows that the 2 hour passage frequency that we used provides a comparable behaviour to chemostat dynamics.

Specific Comments

1. **Page 1, line 18 - Introduction:**

Replace "has enabled" by "have enabled".

corrected

2. **Page 1, line 19 - Introduction:**

The authors may want to clarify what "enabling the construction of chromosome scale, synthetic biological system" means in this context. Is the sentence referring to the current capability of engineering genomes, or synthesising large pieces of DNA, or constructing large constructs? Since the paper does not present a circuit that is integrated into the genome, this sentence can spread confusion unless you have plans of integrating your circuit into the genome, which should perhaps be mentioned later.

We meant that we have the technology to produce very large synthetic constructs. We have clarified this.

3. **Page 2, line 60-61 - Results - Bacteriocins enable population control at a distance:**

Could the authors provide more explanation as to what "strong" and "weak" means in this context? Would "fast growing" and "slow growing" be more suitable in this context?

We have changed to "faster growing" and "slower growing" for clarity.

Could the authors justify their choice of using the JW3910 strain? Would it be relevant to mention that the JW3910 strains is a metB knockout? Given that the strain is grown in M9 media supplemented with casamino acids, did the use of JW3910 decrease the growth rate compared to the wild-type Keio strain?

The JW3910 strain was chosen so that, if necessary, we could alter its growth rate through the use of drop out media as our initial modelling indicated that relative growth rates was a parameter that would allow some control over the community. However, the growth rate difference between the JW3910 engineered strain and the MG1655 competitor strain, when grown in M9 with casamino acids, was enough to produce the competitive exclusion that we required. As such, the added complication of media drop outs was not experimentally explored. We have not compared growth rates to the wild-type Keio strain, but JW3910 does grow slower than the MG1655 used as a competitor in this study, as can be seen from the growth data in Supplementary Figures 1 and 2.

4. **Page 2, line 62-63 - Results - Bacteriocins enable population control at a distance:**

The authors may want to rephrase this sentence to make it more clear that they are investigating a way to reduce the growth rate of the engineered strain by expressing mCherry in the cell. Given that mCherry is expressed from the genome, did the authors notice a decrease in growth rate compared to the strain that is not expressing mCherry? Could these data be provided as supplementary information?

We have clarified this to confirm that the mCherry was expressed from one of the plasmids. We did not ascertain the degree to which the mCherry reduces growth.

4. ****Page 3, line 84-85/Figure 3B - Results - Restricting growth favours bacteriocin producers over fast growers:****

Could the authors provide more explanation as to why serially diluting cultures emulates growth in a chemostat? Intuitively doing serial dilution will change the consortium initial density, as shown in Figure 3B. However that would imply that the cells starting from a smaller density should remain in lagphase for longer than consortia starting with a higher density. Figure 3B does not show this extended lagphase for low starting densities. Could the authors explain why that might be the case?

As the reviewers correctly identifies, the serial dilution changes the initial population density. We disagree that the more dilute cells will remain in lag phase longer; instead the exponential phase will be prolonged, as shown in the figure below. As such, over a given period, the average culture density

and concentrations of bacteriocin and 3OC6-HSL will be dependent on the initial density.

The link between initial density and dilution rate is approximate but the dynamics during exponential phase at a given density in the plate reader should be similar to those in a chemostat with a dilution rate that produces that density. The dilution and initial density both effect the amount of nutrient and other molecules in the media in the same manner, with the pre-dilution growth phase in the experiment designed to condition the media. We have added Supplementary Figure 6 to compare, computationally, the dynamics of our system when simulated in a chemostat with constant dilution versus in repeated batch with different passage periods. The simulations show that for a passage period of 2 hours or below, the batch experiment closely compares to the chemostat experiment.

6. ****Page 5, line 145-146 - Results - Fully autonomous community regulation:****

Could the authors provide more detail as to why they are working with a degradation tag on the C6-HSL synthase (LuxI-LAA)? Could removing the tag improve the production rate of C6-HSL as the synthase would not be degrading so quickly anymore?

The tag was initially used so that arabinose could be used dynamically to control AHL synthesis rate. This was not pursued in this manuscript. It is reasonable to expect that removing the tag would improve the production rate and would allow us to use lower concentrations of arabinose.

7. ****Page 7, line 184 - Results - Identifying robust models for consortia control:****

The McCardell paper presents an architecture to achieve co-existence, but has not demonstrated experimentally that they could achieve co-existence of their bacterial strains. Do the authors refer to the architecture or the results of this paper?

We use the McCardell architecture and explore what dynamics it could theoretically achieve by simulation.

8. ****Page 7, line 193-196 - Discussion:****

The authors claim that their batch experiments carried in microplates follow the results of their in-silico simulations. However, the stable ratio that they observe experimentally seem to happen during the stationary phase during which cells do not divide anymore, hence leading to a stabilised ratio which does not demonstrate that the same would happen in a continuous culture. Without an experiment presenting results maintaining cell consortia ratios in exponential phase as in the presented simulations, there is not enough experimental evidence that the proposed ratiometric control architecture works in vivo.

See our response above to this reviewer's major comment. In summary, the experiments presented show population stability for approximately 2 hours before stationary phase is reached. Further, we have added new experimental data with repeated, periodic dilution of co-cultures that show population stability for approximately 18 hours.

9. ****Page 7, line 205-206 - Discussion:****

One could argue that all previous systems aiming to engineer co-cultures have exploited the principle of competitive exclusion as they all, in their own way (e.g. syntrophic co-cultures), created systems that push strains to rely on the existence of the other to survive in order to modulate their respective growth rates in the co-culture. The authors might want to discuss this a bit more.

Although competitive exclusion is present in all co-cultures, we would argue that the other community control systems work in spite of competitive exclusion whereas our system works because of it; we explicitly require the competitor to grow faster than the engineered strain. The upgrades suggested in Figure 9, remove the requirement for competitive exclusion by introducing a self-limitation motif. We have previously demonstrated that these motifs can improve robustness of community control (Karkaria et al., *bioRxiv* 2020, accepted *Nat. Comms.*).

10. ****Page 7, line 206-210 - Discussion:****

The authors should highlight that the growth context for these two types of applications (engineering the microbiome and production in bioreactors) are very different and require different circuit properties. Have the authors explored (through simulations) whether their circuit has the potential to work in stationary phase, as would be the case in the context of the gut?

We have added some discussion on the functioning of our system under different dilution regimes and related them to the various environments in which one may want to use this system. Our modelling suggests that the system should function at very low dilution rates, and therefore very low resource input rates, as long as the leakiness of the bacteriocin expression is very low. Further, our new Supplementary Figure 6 shows that periodic dilution, as may be found in the context of the gut, can produce dynamics that correspond to constant dilution, depending on the dilution period.

Since gene expression is altered in stationary phase, and our model is characterised with expression in exponential phase, we are unable to use the model to describe the dynamics during stationary phase. We believe it is one of the next big challenges for synthetic biology, to understand and engineer circuits that can function predictably during different growth phases.

11. **Page 8, line 228 - Methods - Strains and Plasmids:**

Neither the plasmid maps nor the plasmid sequences have been added to the supplementary file provided. Will they be provided in a separate file later on?

These have now been uploaded to Zenodo <https://doi.org/10.5281/zenodo.4419900>.

12. **Page 9, line 260 - Methods - Deviations from the above:**

There seems to be a typo with the OD reported. It currently says "01. OD_{700}". This should probably be either 0.1 OD or 0.01 OD.

Corrected

13. **Page 10, line 274-275 - Methods - Agar Plate Spot Inhibition Assay:**

The authors should consider describing which culture (engineered or competitor) they are referring to when they write "the overnight cultures for sampling".

This has been added

14. **Figure 1B:**

Please provide a legend for the black curve? (E+C)

This has been amended.

Please mention explicitly in the legend that the data were obtained experimentally (and not by simulation). Are the data provided the mean of biological replicates? If so, please include error bars. If biological replicates were too noisy, please show the replicates data in the supplementary material.

We have updated the figure legend to state that the data is from plate reader measurements and that each panel includes two replicates, with each point showing the value of an individual replicate.

15. **Figure 2:**

Were biological replicates performed for both panels A and B? If so, please indicate this in the legend.

We have updated the figure legend to state that three replicates were performed, the data shows the mean and standard deviations, apart from Fig2C which just shows the mean of three replicates (as it is not possible to show the error on this type of plot).

16. **Figure 4C:**

Could the authors confirm that the image of the agar plate spot inhibition assay has been inverted when imaging with the GelDoc? It is currently counter-intuitive that the inhibition zone is dark while we would expect to see a "hole" (so lighter) in the competitor strain lawn.

The imager used an epi-white light source which illuminated the plate from above rather than from below. This resulted in the original image shown in Supplementary Figure 3. The image was not inverted.

17. **Figure 5:**

Please provide information about the biological replicates in the legend of Figure 5 as well.

We have updated the figure legend to state that one replicate is shown and a second is in Supplementary Figure 2. Due to the preparation time for this experiment, growth of the cultures after dilution to a given turbidity, means that initial co-culture ratios and densities vary between replicates. The second replicate, included in the supplementary, shows that the results are qualitatively comparable but we believe that the differences in initial conditions are too large for the data to be included in the main figure as a replicate.

18. **Figure 7C:**

Please display the OD for each of the experimental graphs. Can you claim that the co-culture is stable if it is in stationary phase? I recommend the authors provide a complementary experiment that keeps the cells in exponential phase over time. (if a chemostat chamber isn't available to the authors, an experiment performed by manually diluting the cultures before they reach stationary phase so as to keep the cells in exponential growth would provide information about the ability of the proposed ratiometric control system to work as intended)

See our response above to this reviewer's major comment. In summary, the experiments presented show population stability for approximately 2 hours before stationary phase is reached. Further, we have added new experimental data with repeated, periodic dilution of co-cultures that show population stability for approximately 18 hours.

Reviewers' Comments:

Reviewer #1:

Remarks to the Author:

a) The definition of what constitutes a “stable” community.

It appears that the authors use the word “stabilized” to describe a community where competitive exclusion does not happen anymore, i.e. a community that allows for the co-existence of the two focal strains. In ecological theory, however, stability is most often used in the context of resilience, i.e. communities that return to an equilibrium after significant perturbation (e.g. reaching equilibrium, then near-complete removal of one strain, and then return to equilibrium). The “stability” shown in e.g. Fig. 7C appears to only show one “cycle” where competitive exclusion kicks in, the engineered strain makes a bit of bacteriocin and then reaches a balance with the competitor. But what is lacking here is the crucial step of disrupting the system and see if the community can “bounce back” to equilibrium, to fulfil the most commonly used definition of stability. Given how the system is set up I actually expect these simple communities to be formally stable, but I would like to point out that this was technically not shown in this study, so the authors might want to reconsider some of the semantics here.

We use the term “stable” here to discriminate between co-existence and stable population equilibria, the latter being what we aim to achieve. We have included extra experimental data in, what is now, Figure 8 in which the co-cultures are repeatedly diluted every 2 hours for 18 hours. Although this is intended to approximate chemostat dilution, it provides a repeated periodic disruption to the community, as suggested by the reviewer, and confirms the stability of the cocultures.

- I appreciate that the authors generated an additional dataset that looks at strain co-existence after repeated a mild forms of perturbation, i.e. repeated dilution. While this does support their claim that the system could be formally “stable” - at least over 18h - I am still missing a harsher perturbation such as e.g. the near-complete removal of one strain after the system has stabilized once, and test if the system returns to equilibrium. Also, and as other referees have also pointed out, the strongest case of stability can be made using chemostat experiments where nutrients are never depleted, but this is still lacking. See also my ongoing concerns about describing the system as “fully autonomous” (see below).

b) The abstract stating that the study considers “multi-species”(L12) and “two-species consortia” (L16). The presented experiments cover a pairwise interaction between two strains of the same species (E. coli), so the abstract should be rephrased to reflect this.

We have changed the abstract to reflect that a two-strain community was used.

- Ok.

c) The limitations of using bacteriocins. Most bacteriocins are extremely narrow in their target range and can only target strains within the same species, or sometimes closely related species. The use of a bacteriocin (microcin V) in this system thus severely limits its potential for the (discussed) purpose of controlling natural microbiota. Realistically, you can probably only control one strain at a time (or maybe a really closely related species like Salmonella, if we are being generous). So, I would argue that the system described here - using an engineered strain producing a single, narrow-range bacteriocin - is not as broadly applicable as the authors make it out to be, and it would not allow control of "a community through a single constituent" (L47) if we consider naturally occurring communities (e.g. the microbiome) which are made up of hundreds of species. In L206, the authors state that “this allows the construction of synthetic communities while only requiring the engineering of a single strain, a feature that enables applications in which an engineered population can coexist with native microbiomes.”, except the single engineered strain only kills a very specific type of competitor (the one with the specific bacteriocin receptor on its membrane), so application is limited to "communities" of E. coli and relatives, and not to actual mixed-species native microbiomes. I suggest the authors discuss this caveat more explicitly in the introduction and discussion.

We have added further context to the introduction regarding the spectrum of bacteriocin activity.

We have highlighted that, although most bacteriocins seem to show a narrow spectrum of activity, some do exist with a broader spectrum. We also highlight that there have been successful attempts to engineer bacteriocins to change their targets and increase their spectrum and that many species express multiple bacteriocins. We agree that the system demonstrated here is limited in its ability to broadly act against a microbiome. However, coexistence may be achieved by targeting a single, abundant, strain in the microbiome for niche creation. To this effect, the bacteriocin that we used can be exchanged for another, or indeed combined with others, in order to target another niche.

- Ok.

d) Describing the system as “fully autonomous”. The authors postulate that their system does not need “environmental control” to achieve stability, but I would say adding an external stimulant such as a high concentration of arabinose (as shown in Fig. 7C) is environmental control. It basically means that parts of the system need to be cranked up to full power (maximally induced production of LuxI, which regulates the production of QS molecules) by an externally added chemical stimulus to achieve strain co-existence. I would argue that the “true” benchmark of an autonomously self-regulated system is one that can be perturbed (see comment on “stability” above) and will return to a pre-defined stable state without external addition of any direct stimulant of the system.

The use of arabinose for induction of LuxI enabled us to find a region of LuxI expression which allows coexistence. This could have been achieved with constitutive promoters but would have involved extra cloning for every expression level modification. Further, although the environment is established to illicit a particular response, there is no external interaction with the environment other than the setup. As such, we believe it is valid to describe the system as fully autonomous. We have added some clarification to the discussion.

- While I understand the reasoning, I still don't think the label “fully autonomous” is warranted. I agree that after the initial setup there is no external stimulus needed anymore, but if we think about using these types of systems in communities like e.g. the microbiome, it is highly unlikely that we will ever be able to ensure the presence of a given stimulating chemical at exactly the right concentration for a given strain-strain interaction to be stable. In fact, I am hoping that moving forward, “better” versions of this system will not have to rely on externally added chemicals anymore to achieve co-existence. I therefore think the “high standard” of labelling something as fully autonomous should be reserved for systems that go beyond what is presented here in terms of independence from exogenously added chemicals.

In summary, I think this study covers an interesting topic, the experiments are well described in the methods and the analyses are appropriate. I think the introduction and discussion could benefit from more fully addressing the obvious limitations of this system for actual multi-species community control, and the use of labels like “stability” and “full autonomy” should be considered more carefully throughout the manuscript.

Minor comments:

1. L67 (whole paragraph); what is the rationale for the starting ratios of 75:25 here? Competition dynamics in this system are highly frequency-dependent (as the authors show nicely in Fig. 2C), so it would be helpful if starting ratios were always explicitly mentioned when describing competition outcomes in the results section.

We have added to the text that this particular timeseries was chosen as it most clearly showed competitive exclusion followed by bacteriocin killing. The timeseries was chosen *post hoc* from the data shown in Supplementary Figure 1. In principle several other timeseries could have been used but the period of competitive exclusion would be compressed and more difficult to see.

- Ok.

2. L73-74; I am a bit confused about this statement. How does this follow from the results in Fig. 2C? Elaborating on this could help the reader follow the logic a bit better.

We have removed this statement as it was a pre-emptive rewording of the following section.

- Ok.

3. L124 “use population density for control”; adding arabinose forces them to produce large amounts of quorum molecules per cell, so how does this interfere with them actually sensing density? Does this simply lower the QS threshold, so that they switch off bacteriocin production faster/earlier during the competition? The interaction between arabinose induction and sensing density could be explained a bit more in detail here.

It is correct that increasing arabinose induction, increases the production quorum molecules per cell and therefore reduces the population threshold at which the bacteriocin turns off. We have extended the explanation as suggested.

- Ok.

4. L158 typo; the viability [of] strains?

Corrected.

- Ok.

5. L196 “first time ... dose-dependent expression of bacteriocins”. Colicins are routinely expressed in a dose-dependent way by addition of mitomycin C or other DNA-damaging agents (e.g. <https://journals.plos.org/plosone/article?id=10.1371/journal.pone.0119124>). Perhaps this statement should be restricted to microcins if dose-dependent induction was indeed never shown before for them. We thank the reviewer for bringing this literature to our attention. We have removed the “first time” from our statement.

- Ok.

6. L269; shouldn't this be strain JW3910? MG1655 is the fast grower non-producer I thought? This assay was performed in MG1655 transformed with the relevant plasmids, before we decided to switch to JW3910 as the host strain. As this assay does not rely on relative growth dynamics, we believe it is reasonable to assume that the results can be extrapolated to the JW3910. The change of host strains was due to the modelling demonstrating that growth rate differences between engineered and competitor strains could be used to tune competition (which we could theoretically do using the auxotrophy), however this was not experimentally explored.

- Ok.

7. L275 typo “minutes”

Corrected.

- Ok.

8. Figures – what is the difference between the experiments shown in Fig. 1B and those shown in Fig. 2? Adding description of experiments to Fig.1 legend would help here.

Experimental differences are highlighted in the “Deviations from the above” subsection in the “plate reader competition assays” section of the Methods. Specifically, Figure 1B was performed with a slightly higher volume, a different plate sealing technology and a different measurement frequency, than Figure 2.

- Ok.

9. Fig. 2C, the y axis is a bit confusing here – are those discrete fractions? So we are looking at four experiments? But the colour blocks don't correspond (or line up well to) the y axis ticks?

Each horizontal represents a single timecourse, so there are four timecourses as the reviewer notes. The blocks are horizontally centred on the initial co-culture ratio. For clarity, we have changed the axis ticks to reflect these initial ratios.

- Ok.

Reviewer #2:

Remarks to the Author:

In the revised manuscript, the authors addressed a few of my previous concerns. However, the major concerns relating to the advantages and utility of the proposed approach remain.

1) Achieving population coexistence by killing fast-growing cells is a heavy waste of energy and resources, which the authors also agreed. To justify the benefits of their approach, the authors argued that there are cases "where we wish to engineer niche creation, there may be no other choice". It is plausible in principle but to make a strong case, the authors shall demonstrate it through a direct experiment, rather than verbal arguments in introduction. The current manuscript only shows the control of one strain to the other, hard to justify the use of the approach to manipulate an existing ecosystem.

2) I disagree with the authors' argument that engineering multiple species of an ecosystem is a disadvantage of current synthetic ecosystems. By dividing the tasks of genetic engineering into multiple species, engineering multiple species allows compartmentalization and separate construction, which is much simpler and more scalable than targeting on a single strain. There could be a regime where increasing the number of engineered species will increase system complexity. However, it is true even for the engineering of single strains – more parts you introduce into a single strain, more complex and challenge to implement. Additionally, placing the same amounts of parts into a single species is much more challenging than distributing them into multiple strains.

Reviewer #3:

Remarks to the Author:

The authors have addressed our major comments and have gathered new compelling experimental data to show that the ratio of their co-cultures can stabilize in chemostat environments.

This reviewer is quite curious why in the batch experiment, the proportion of E cells settles to ~ 0.5 but in their repeated dilution experiment (simulating a chemostat environment), the proportion of E cells settles at half that value (~ 0.25). The authors haven't discussed why this difference might arise. I think explaining this would be interesting and useful.

REVIEWER COMMENTS

Reviewer #1 (Remarks to the Author):

[The definition of what constitutes a “stable” community.]

I appreciate that the authors generated an additional dataset that looks at strain co-existence after repeated a mild forms of perturbation, i.e. repeated dilution. While this does support their claim that the system could be formally “stable” - at least over 18h - I am still missing a harsher perturbation such as e.g. the near-complete removal of one strain after the system has stabilized once, and test if the system returns to equilibrium.

We understand this comment however, again it comes down to a question of what you define as stable. What the reviewer describes is a system that is more like Lyapunov stable, that is a structural stable system. We did not claim we had created such as system, we created a system with a locally stable equilibrium. The experiments we performed clearly show that this system is stable up to relatively large perturbations. A structurally stable system would be a very interesting and useful future goal for this type of approach.

Also, and as other referees have also pointed out, the strongest case of stability can be made using chemostat experiments where nutrients are never depleted, but this is still lacking. See also my ongoing concerns about describing the system as “fully autonomous” (see below).

Even in a chemostat nutrients are depleted, and a carrying capacity reached based on the rate of dilution and consumption by the resident strains. The cultures in our repeated dilution experiments are maintained in exponential growth phase, well below the batch carrying capacity, as can be seen in Figure 8E.

[Describing the system as “fully autonomous”.]

While I understand the reasoning, I still don't think the label “fully autonomous” is warranted. I agree that after the initial setup there is no external stimulus needed anymore, but if we think about using these types of systems in communities like e.g. the microbiome, it is highly unlikely that we will ever be able to ensure the presence of a given stimulating chemical at exactly the right concentration for a given strain-strain interaction to be stable. In fact, I am hoping that moving forward, “better” versions of this system will not have to rely on externally added chemicals anymore to achieve co-existence. I therefore think the “high standard” of labelling something as fully autonomous should be reserved for systems that go beyond what is presented here in terms of independence from exogenously added chemicals.

We understand the reviewer's argument. If we changed the promoter for a constitutive one, such that arabinose is no longer part of the system, then we would have a fully autonomous system. Since we haven't done this yet, we have changed “fully autonomous” to “autonomous”.

Reviewer #2 (Remarks to the Author):

In the revised manuscript, the authors addressed a few of my previous concerns. However, the major concerns relating to the advantages and utility of the proposed approach remain.

1) Achieving population coexistence by killing fast-growing cells is a heavy waste of energy and resources, which the authors also agreed. To justify the benefits of their approach, the authors

argued that there are cases "where we wish to engineer niche creation, there may be no other choice". It is plausible in principle but to make a strong case, the authors shall demonstrate it through a direct experiment, rather than verbal arguments in introduction. The current manuscript only shows the control of one strain to the other, hard to justify the use of the approach to manipulate an existing ecosystem.

We would argue that the whole paper is demonstrating niche creation. Figure 1B shows how a niche is created by our population control system. In the system that doesn't produce bacteriocin, no niche is created, and the engineered strain is outcompeted. When bacteriocin is produced, a niche is created, where the engineered strain is no longer outcompeted. The rest of the paper deals with how to tune and balance growth vs competition. To demonstrate the principle, and aid the mathematical modelling, we have used a monoculture as the background system. We also argue that this represents a valid use case (see argument below). However, we take the reviewers comment on board and have added the following into the discussion:

"We have engineered a proof-of-principle population control system using a monoculture as the native community in order to establish a niche. Future work will involve implementation of the system in more complex communities [48,49] and in animal models including C. elegans [50]."

2) I disagree with the authors' argument that engineering multiple species of an ecosystem is a disadvantage of current synthetic ecosystems. By dividing the tasks of genetic engineering into multiple species, engineering multiple species allows compartmentalization and separate construction, which is much simpler and more scalable than targeting on a single strain. There could be a regime where increasing the number of engineered species will increase system complexity. However, it is true even for the engineering of single strains – more parts you introduce into a single strain, more complex and challenge to implement. Additionally, placing the same amounts of parts into a single species is much more challenging than distributing them into multiple strains.

We see three scenarios for microbial communities and synthetic biology; bespoke systems, interchangeable modular toolkits and infiltrating extant microbiota.

The first is the *de novo* engineering of a community in which all constituent members are designed from the beginning to be part of the community. In these circumstances we can choose whichever method we desire for community control and therefore our method of control using killing does provide an alternative tool to consider when designing communities. However, having said that, it is commonly understood that biological engineering needs to move away from such bespoke designs if it is to have the impact that standardisation affords, for example as has happened for most engineering disciplines.

In the second use, the community is made up of "off-the-shelf" engineered strains that have not been designed to be part of a specific community but can be assembled to form one. In this setting, although it would be possible to re-engineer the strains to incorporate community control, this would likely affect the pre-characterised function of those strains. As such, we believe that a specialised community control strain would be of use to enable the predictable assembly of the community. Indeed, this would be an example of the compartmentalisation of function that the reviewer highlights above. In this manuscript, the system that we present provides the first steps

towards that community control strain, though additional bacteriocins to target more community members will be necessary.

In the final scenario, rather than the assembly of a new community, the goal is to infiltrate an engineered strain into an existing microbiome. In this scenario there is no possibility to engineer any of the constituent members. This is the setting for the niche creation behaviour that the reviewer refers to in their comment above. The microbial systems that we wish to infiltrate will vary greatly in complexity. If we wish to maintain a live therapeutic in the gut, the resident microbiota is incredibly diverse but is still dominated by particular taxa (Huttenhower et al., Nature, 2012) which can be targeted by bacteriocins. Alternatively, we may wish to introduce our strain into far simpler systems, for example as biosensors in bioreactors, which may comprise a single resident strain. In all these settings, the ability of the niche creation system to limit its impact on the resident microbiota is key.

Reviewer #3 (Remarks to the Author):

The authors have addressed our major comments and have gathered new compelling experimental data to show that the ratio of their co-cultures can stabilize in chemostat environments.

This reviewer is quite curious why in the batch experiment, the proportion of E cells settles to ~ 0.5 but in their repeated dilution experiment (simulating a chemostat environment), the proportion of E cells settles at half that value (~ 0.25). The authors haven't discussed why this difference might arise. I think explaining this would be interesting and useful.

Yes this is interesting. In fact this is predicted from the modelling (Supplementary Figure 6). In the batch experiment we use the sum of behaviour over a 5 hour time period to approximate behaviour in the chemostat. This is equivalent to having a 5 hour dilution period in the repeated dilution experiment. Supplementary Figure 6 shows that the 5 hour period results in a shift in the region of coexistence which leads to a higher proportion of E in the population, compared to the repeated dilution with a 2 hour period, while estimated dilution rate and arabinose concentration are the same. We have added a statement about this in the main text:

"We see a shift in the equilibrium population ratios between the batch experiment (Figure 8A) and the repeated dilution experiment (Figure 8D). This is predicted from the modelling (Supplementary Figure 6) as, in our batch experiments, the period that we use to approximate chemostat behaviour is 5 hours. This results in a shift in the region of co-existence, leading to a higher equilibrium proportion of engineered strain for the same dilution rate and arabinose concentration."